# HUWE1 controls tristetraprolin proteasomal degradation by regulating its phosphorylation

Sara Scinicariello[1,2], Adrian Soderholm[1,2], Markus Schäfer[3], Alexandra Shulkina[1,2], Irene Schwartz[1,2], Kathrin Hacker[1], Rebeca Gogova[2,3], Robert Kalis[2,3], Kimon Froussios[3], Valentina Budroni[1,2], Annika Bestehorn[1,2], Tim Clausen[3,4], Pavel Kovarik[1], Johannes Zuber[3,4], Gijs A Versteeg[1]*

[1]Department of Microbiology, Immunobiology and Genetics, Max Perutz Labs, University of Vienna, Vienna BioCenter (VBC), Vienna, Austria; [2]Vienna BioCenter PhD Program, Doctoral School of the University of Vienna and Medical University of Vienna, Vienna BioCenter (VBC), Vienna, Austria; [3]Research Institute of Molecular Pathology (IMP), Vienna BioCenter (VBC), Vienna, Austria; [4]Medical University of Vienna, Vienna BioCenter (VBC), Vienna, Austria

*For correspondence:
gijs.versteeg@univie.ac.at

Competing interest: The authors declare that no competing interests exist.

**Abstract** Tristetraprolin (TTP) is a critical negative immune regulator. It binds AU-rich elements in the untranslated-regions of many mRNAs encoding pro-inflammatory mediators, thereby accelerating their decay. A key but poorly understood mechanism of TTP regulation is its timely proteolytic removal: TTP is degraded by the proteasome through yet unidentified phosphorylation-controlled drivers. In this study, we set out to identify factors controlling TTP stability. Cellular assays showed that TTP is strongly lysine-ubiquitinated, which is required for its turnover. A genetic screen identified the ubiquitin E3 ligase HUWE1 as a strong regulator of TTP proteasomal degradation, which we found to control TTP stability indirectly by regulating its phosphorylation. Pharmacological assessment of multiple kinases revealed that HUWE1-regulated TTP phosphorylation and stability was independent of the previously characterized effects of MAPK-mediated S52/S178 phosphorylation. HUWE1 function was dependent on phosphatase and E3 ligase binding sites identified in the TTP C-terminus. Our findings indicate that while phosphorylation of S52/S178 is critical for TTP stabilization at earlier times after pro-inflammatory stimulation, phosphorylation of the TTP C-terminus controls its stability at later stages.

## Editor's evaluation

The study by Scinicariello et al. set out to identify novel factors that controlled TTP stability and identified HUWE1 by CRISPR screening in macrophages. HUWE1 limited TTP phosphorylation at later times post inflammatory stimulation on residues distinct from those functionally modified by MAPKs in early stages. Overall, the biochemical and cellular signaling experiments were thoughtfully designed and well executed, leading to the discovery of HUWE1 as a TTP regulator.

## Introduction

Dynamic regulation of the immune system is essential to mount a defense against pathogens upon infection, yet shut-off the response at the appropriate time during resolution. Since most cytokines and other pro-inflammatory mediators are transcriptionally induced during infection, an essential aspect of returning to homeostatic conditions is the timely removal of their mRNAs during resolution.

Tristetraprolin (TTP; also known as ZFP36 or TIS11A) is an RNA-binding protein that interacts with AU-rich elements (ARE) present in the 3'-untranslated-regions (UTR) of many mRNAs encoding pro-inflammatory mediators (*Galloway et al., 2016*; *Lai et al., 1999*; *Sedlyarov et al., 2016*). Subsequently, TTP recruits the CCR4-NOT decapping and deadenylation complex to target mRNAs, resulting in their destabilization and removal from the cell (*Fabian et al., 2013*; *Lai et al., 2003*; *Lykke-Andersen and Wagner, 2005*; *Sandler et al., 2011*; *Tiedje et al., 2016*). TTP binds to the AREs of a multitude of mRNAs encoding cytokines and other immune-related factors, yet not all of them are destabilized (*Moore et al., 2018*; *Sedlyarov et al., 2016*; *Tiedje et al., 2016*; *Zhang et al., 2017*). This has suggested that additional -hitherto unknown- regulatory mechanisms are at play controlling TTP-dependent mRNA degradation, which may differ in various cell types.

The biological importance of TTP for proper dampening of the inflammatory response is underpinned by the observation that *Zfp36* (the gene encoding TTP)-deficient mice develop systemic inflammation characterized by arthritis, dermatitis, conjunctivitis, and cachexia, which has been coined TTP deficiency syndrome (*Taylor et al., 1996*). One of the main deregulated ARE-containing mRNAs driving the inflammatory phenotype in *Zfp36*-deficient mice is *Tnf* (*Carballo et al., 1998*; *Taylor et al., 1996*), although additionally *Il1a/b*, *Il23*, and *Ccl3* have been implicated as well (*Kang et al., 2011*; *Molle et al., 2013*; *Sneezum et al., 2020*).

TTP itself is regulated at the transcriptional, post-transcriptional, and post-translational levels. Most cell types express low levels of *Zfp36* mRNA in unstimulated conditions, the transcription of which is robustly induced by proinflammatory stimuli including the Toll-like receptor 4 (TLR4) agonist lipopolysaccharide (LPS) in myeloid cells such as macrophages (*Carballo et al., 1998*; *Lai et al., 1995*; *Sauer et al., 2006*; *Schaljo et al., 2009*; *Suzuki et al., 2003*). At the post-translational level, TTP is phosphorylated at over 30 residues by inflammation-activated stress kinases (*Brook et al., 2006*; *Clark and Dean, 2016*; *Hitti et al., 2006*; *Ronkina et al., 2019*).

The biological relevance of most TTP phospho-sites and the identity of the involved kinases remain unknown (*Clark and Dean, 2016*; *Ronkina et al., 2019*). Most characterized are phosphorylation events at residues S52 and S178 in murine TTP that are mediated by the inflammation-activated kinase MK2, which acts down-stream of p38 mitogen-activated protein kinase (MAPK; *Brook et al., 2006*; *Hitti et al., 2006*). In mice, TTP mutants lacking these phosphorylation sites are highly unstable and rapidly degraded, yet highly biologically active (*Ross et al., 2015*).

This has given rise to a model in which TTP is predominantly unphosphorylated and rapidly degraded in unstimulated cells, whereas pro-inflammatory cell signaling not only increases *Zfp36* transcription, but also TTP S52/S178 phosphorylation and stabilization through interaction with 14-3-3 proteins (*Kratochvill et al., 2011*; *Sedlyarov et al., 2016*). However, in this S52/S178 phosphorylated state, TTP is thought to be inactive, whereas during dephosphorylation of these residues at later times in the inflammatory response, TTP actively mediates mRNA degradation (*Kratochvill et al., 2011*; *Sedlyarov et al., 2016*). Nevertheless, the impact of the other 30+ phosphorylated residues on TTP stability and activity has remained largely elusive.

Proteasomes are the main degradation machines of cells for homeostatic protein turn-over (*Bard et al., 2018*). 20S core particles contain catalytic activity, yet lack receptors for ubiquitin and ATPase activity for unfolding and translocation of proteins into the catalytic chamber. Association of 19S regulatory particles containing ubiquitin receptors and AAA+ ATPase activity assembles 26S proteasomes, which are considered the main degradative entities for poly-ubiquitinated proteins (*Bard et al., 2018*).

TTP protein is degraded through the proteasome, as previous studies showed that 20S proteasome inhibition stabilizes TTP. Moreover, a previous study suggested that TTP may be directly degraded by 20S proteasomes in a ubiquitin-independent manner. In this context, important destabilizing intrinsically disordered regions in the N and C termini of TTP were identified and have been suggested to putatively allow direct degradation by 20S proteasomes (*Brook et al., 2006*; *Ngoc et al., 2014*). Yet, other regulators of intracellular TTP protein abundance have remained elusive. In this study, we set out to identify and characterize novel factors that control TTP turn-over, thereby affecting pro-inflammatory output.

Through genetic loss-of-function screening, we identified several novel determinants of TTP abundance, including the giant ubiquitin E3 ligase HUWE1. Our data indicate that TTP is strongly poly-ubiquitinated on lysines in its zinc finger domain, and degraded by the proteasome in a ubiquitin-dependent manner. Moreover, we identified a novel role for the E3 ligase HUWE1 in indirectly

controlling TTP turn-over through mediating its phosphorylation via multiple stress kinases, and reduced dephosphorylation.

## Results

### TTP is degraded in a ubiquitin-dependent manner

Pro-inflammatory stimuli such as LPS drive both transcription of *Zfp36* (the gene encoding TTP), and phosphorylation of the TTP protein. To study how TTP protein levels are regulated, we established a macrophage cell line expressing exogenous TTP from a constitutively active promoter, uncoupling *Zfp36* transcription from regulatory effects on TTP protein stability in the absence or presence of pro-inflammatory signals.

Consistent with previous studies, endogenous TTP protein was rapidly induced by LPS (*Carballo et al., 1998*; *Lai et al., 1995*; *Sauer et al., 2006*; *Schaljo et al., 2009*; *Suzuki et al., 2003*), and in the absence of de novo protein synthesis, rapidly degraded (*Figure 1A*; *Brook et al., 2006*; *Ngoc et al., 2014*). Treatment with proteasome inhibitor MG132 almost completely prevented TTP degradation, indicating that its degradation is predominantly through the proteasome. Under these conditions, a high-MW form of TTP accumulated, suggesting that phosphorylation is important for regulation of its stability.

To investigate whether pro-inflammatory stimuli are exclusively stabilizing TTP, or also provide degradation signals, a macrophage cell line stably expressing HA-tagged TTP was established (*Figure 1B*). Under non-stimulated conditions, HA-TTP was detected as medium-range MW species migrating at and above its predicted MW of 36.7 kDa, consistent with it being partially phosphorylated under non-stimulated conditions. Stimulation with LPS resulted in rapid TTP stabilization after 30 min, followed by a reduction of its protein levels at 3 hr and 7 hr post-treatment (*Figure 1B*; lanes 2–3). This suggested that pro-inflammatory stimuli may also provide the signaling required for TTP turn-over at longer stimulation times, possibly through regulating its phosphorylation.

To determine whether TTP proteasomal degradation was mediated by ubiquitination, cells were treated with the ubiquitin E1 inhibitor TAK-243, which inhibits de novo ubiquitination (*Figure 1B* and *Figure 1—figure supplement 1A–B*). This stabilized endogenous and exogenously expressed TTP under baseline and LPS-stimulated conditions (*Figure 1B*, and *Figure 1—figure supplement 1A–B*), demonstrating that TTP is degraded in a ubiquitination-dependent manner. Consistent with this notion, exogenous HA-TTP and endogenous TTP was detected to be ubiquitinated in denaturing lysates from these cells (*Figure 1C*, and *Figure 1—figure supplement 1C-D*).

Moreover, a TTP mutant in which all of its five lysine residues in the TTP zinc finger domain were mutated to arginines (KtoR), accumulated at high steady-state levels, and was substantially less ubiquitinated (*Figure 1—figure supplement 1C*). Consistent with its strongly reduced ubiquitination, the KtoR TTP mutant was stabilized (*Figure 1—figure supplement 1E*). Mutation of individual lysines had no significant effects on TTP accumulation (*Figure 1—figure supplement 1F*), suggesting that multiple lysine residues in TTP may be functionally redundant for its ubiquitination and degradation. A TTP mutant with simultaneous mutation of four residues (K97/115/133/135 R) did significantly accumulate, albeit to a lesser extent than a mutant in which all five lysines were mutated (*Figure 1—figure supplement 1F*). In line with lysine poly-ubiquitination playing an important role in TTP degradation, degradative K48-linked poly-ubiquitin chains were detected on TTP, whereas non-degradative K63-linked chains were not (*Figure 1—figure supplement 1G*). Collectively, these results indicate that TTP is covalently poly-ubiquitinated in its TTP zinc finger domain, and that all five lysines are functionally important for TTP degradation.

To enable identification of TTP abundance regulators by genetic screening, a macrophage cell line stably expressing unstable mCherry-TTP and stable BFP was established (*Figure 1D*). The stable BFP served as an internal control, as it is translated in equimolar amounts from the same transcript through a P2A ribosomal skip site. mCherry-TTP accumulated in cells as a stable protein under non-stimulated conditions (*Figure 1E*; top panel). In contrast, LPS stimulation initially further stabilized mCherry-TTP, yet subsequently facilitated its degradation, phenocopying its endogenous TTP counterpart (*Figure 1E*; bottom panel, and *Figure 1—figure supplement 1H*). Treatment of lysates from these cells with alkaline phosphatase collapsed higher migrating endogenous and exogenous TTP species

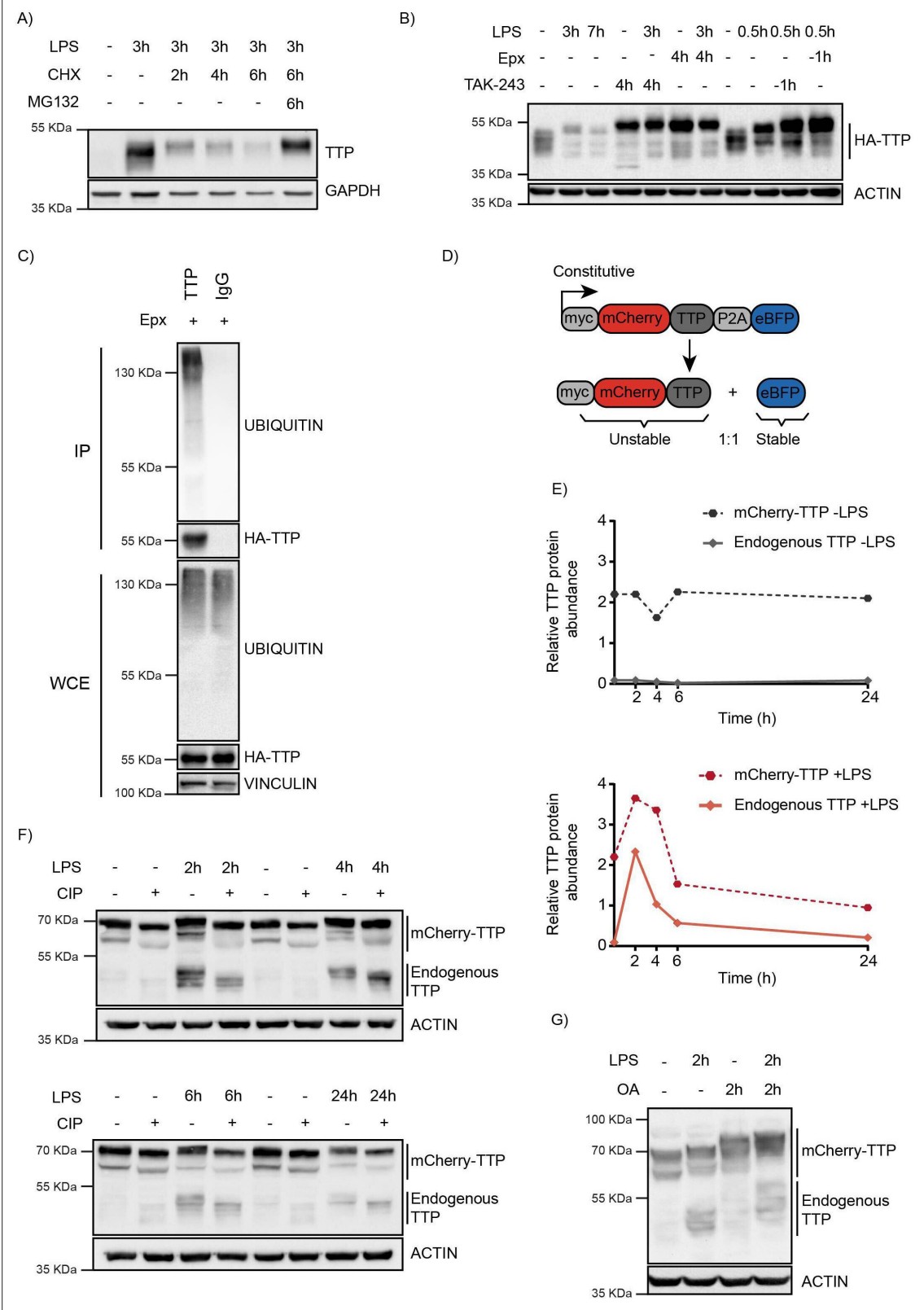

**Figure 1.** TTP is degraded by the proteasome in a ubiquitin-dependent manner. (**A**) RAW264.7 murine macrophages were stimulated with LPS and incubated with the translation inhibitor cycloheximide (CHX) and the proteasome inhibitor MG132 for the indicated times (**h**), after which TTP levels were analyzed by western blot. (**B**) 3xHA-TTP-expressing RAW264.7 cells were incubated with LPS or left unstimulated. Cells were then treated with E1 enzyme inhibitor (TAK-234) or the proteasome inhibitor Epoxomicin (Epx). Protein levels were assessed by western blot. (**C**) RAW264.7 cells stably

*Figure 1 continued on next page*

*Figure 1 continued*

expressing 3xHA-tagged TTP protein were treated with Epx for 5 hr, after which TTP was immunoprecipitated, and its ubiquination analyzed by western blot. (**D**) Schematic representation of the TTP stability reporter construct. Constitutively expressed myc-tagged mCherry-TTP fusion protein and enhanced blue fluorescent protein (eBFP2) are translated at equimolar levels through a P2A site. (**E–F**) RAW264.7-Dox-Cas9-mCherry-TTP cells were stimulated with LPS for the indicated times. Subsequently, cell lysates were treated with Calf Intestinal Phosphatase (CIP) for 2 hr at 37 °C, and TTP levels analyzed by western blot. Non-saturated western blot signals for mCherry-TTP and endogenous TTP protein were quantified, normalized to ACTIN levels, and plotted. (**G**) RAW264.7-Dox-Cas9-mCherry-TTP cells were treated with LPS for 2 hr, after which PP1/2 inhibitor okadaic acid (OA) was added to the culture medium for 2 hr. TTP electrophoretic mobility was assessed by western blot.

The online version of this article includes the following source data and figure supplement(s) for figure 1:

**Source data 1.** Blots corresponding to *Figure 1A*.

**Source data 2.** Blots corresponding to *Figure 1B* and *Figure 1—figure supplement 1A*.

**Source data 3.** Blots corresponding to *Figure 1C*.

**Source data 4.** Blots corresponding to *Figure 1F*.

**Source data 5.** Blots corresponding to *Figure 1G*.

**Figure supplement 1.** TTP is degraded by the proteasome in a ubiquitin-dependent manner.

**Figure supplement 1—source data 1.** Blots corresponding to *Figure 1—figure supplement 1B*.

**Figure supplement 1—source data 2.** Blots corresponding to *Figure 1—figure supplement 1C*.

**Figure supplement 1—source data 3.** Blots corresponding to *Figure 1—figure supplement 1D*.

**Figure supplement 1—source data 4.** Blots corresponding to *Figure 1—figure supplement 1E*.

**Figure supplement 1—source data 5.** Blots corresponding to *Figure 1—figure supplement 1F*.

**Figure supplement 1—source data 6.** Blots corresponding to *Figure 1—figure supplement 1G*.

**Figure supplement 1—source data 7.** Blots corresponding to *Figure 1—figure supplement 1I*.

**Figure supplement 1—source data 8.** TurboID-TTP proximity labelling in RAW264.7 cells.

(*Figure 1F*), whereas inhibition of the phosphatases PP1 and PP2 by okadaic acid (OA) increased them (*Figure 1G*), indicating that mCherry-TTP is phosphorylated in a similar fashion as endogenous TTP.

Together, these data show that LPS-stimulation initially stabilizes TTP, whereas at later time points its induced cell signaling events direct TTP degradation.

## The E3 ligase HUWE1 is a major determinant of cellular TTP protein abundance

Next, we set out to identify cellular factors regulating TTP protein abundance. To this end, a RAW264.7 mouse macrophage cell line with Dox-inducible Cas9 was established, which in addition expresses mCherry-TTP (*Figure 1D* and *Figure 2A*). To enable identification of essential genes, a cell line was established which only functionally edits in the presence of doxycycline (Dox), but not in its absence (*Figure 2—source data 4*).

A genome-wide lentiviral sgRNA library was transduced into these cells, ensuring one integration per cell. Knock-outs were induced by treatment with Dox for three and six days to identify regulators irrespective of essential gene functions and different protein half-lives. Subsequently, TTP was destabilized by LPS treatment, cells with high and low mCherry-TTP content were sorted, and their integrated sgRNA coding sequences quantified by next-generation sequencing (*Figure 2A*, and *Figure 2—source data 5*). In parallel, sorted cells with high or low levels of the stable BFP control (*Figure 1D*, and *Figure 2—source data 5*) were likewise processed, and used for identifying non-specific factors.

As anticipated, factors previously reported to be important for stabilizing TTP (e.g. *Mapkapk2 (Mk2)*, *Ywhag (14-3-3γ)*, *Mapk14 (p38)*) were significantly enriched in the mCherry-TTP^low cell pool (*Figure 2—source data 6*, *Figure 2B* and *Figure 2—figure supplement 1A*; left and top panels, respectively). Consistent with mCherry-TTP proteasomal degradation being LPS-dependent (*Figure 1E*), key factors for TLR4-signaling (*Tlr4*, *Ly96 (Md2)*), and components of the proteasome (*Psma5*, *Psmb7*) were significantly enriched in mCherry-TTP^high sorted cells (*Figure 2B* and *Figure 2—figure supplement 1A*; right and bottom panels, respectively). Moreover, various additional new candidates controlling cellular TTP abundance were identified, including the giant ubiquitin E3 ligase *Huwe1* (*Figure 2B* and *Figure 2—figure supplement 1A*, right and top panels, respectively). Individual targeting of these

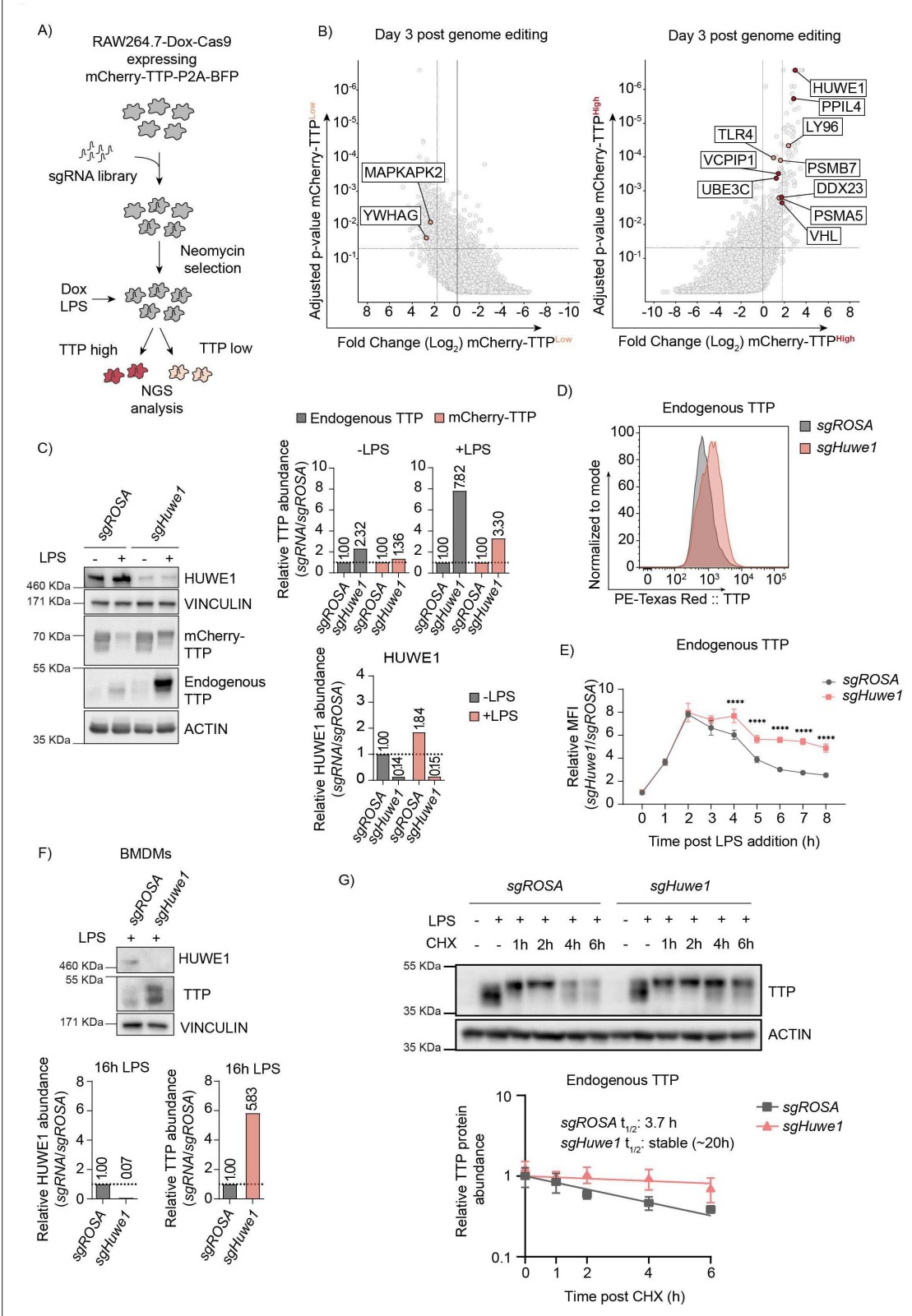

**Figure 2.** Genome-wide CRISPR-Cas9 knockout screen identified the E3 ligase HUWE1 as a regulator of TTP stability. (**A**) Overview of FACS-based CRISPR-Cas9 knockout screening procedure using the RAW264.7-Dox-Cas9-mCherry-TTP cell line. Cells expressing high and low levels of mCherry-TTP protein were sorted, and their integrated sgRNA coding sequences determined by next generation sequencing. (**B**) Read counts per million in the mCherry-TTP^high cells at 3 days after Cas9 induction were compared to those in unsorted cells from the same day, sgRNA enrichment calculated by

*Figure 2 continued on next page*

*Figure 2 continued*

MAGeCK analysis, and log2-fold change and adjusted p-value plotted. Genes enriched in the sorted populations that met the following criteria are indicated in red: a log2 fold-change of <1.8 (mCherry-TTP^low) or >1.8 (mCherry-TTP^high), adjusted p-value <0.05, not enriched in the matching eBFP2^low or eBFP2^high sorted cells. (**C**) Cas9 was induced with Dox for 5 days in RAW264.7-Dox-Cas9-mCherry-TTP cells expressing either sg*ROSA* or sg*Huwe1*. Subsequently, cells were treated with LPS for 16 hr, and TTP protein levels were assessed by western blot. HUWE1, mCherry-TTP and endogenous TTP abundance was quantified and plotted. The TTP and ACTIN panels are the left four lanes from the blot presented in *Figure 2—figure supplement 1B*. (**D**) RAW264.7-Dox-Cas9 cells expressing sg*ROSA* or sg*Huwe1* were treated with Dox for 5 days to induce Cas9. Then, cells were incubated with LPS for 16 hr or left unstimulated, and endogenous TTP protein levels analyzed by intracellular staining, followed by flow cytometry. (**E**) sg*ROSA*- or sg*Huwe1*-targeted RAW264.7-Dox-Cas9 cells were treated with LPS for the indicated times (**h**), and TTP levels were analyzed by flow cytometry. Normalized mean fluorescence intensity (MFI) was plotted. Data represent the mean and s.d.; n=3 biological replicates. ****p ≤0.0001. (**F**) Bone marrow-derived macrophages (BMDMs) isolated from Cas9-expressing knock-in mice were stably transduced with sg*ROSA* or sg*Huwe1*. Cells were incubated with LPS for 16 hr or left unstimulated. Endogenous TTP protein levels were determined by western blot. Quantified TTP levels normalized to VINCULIN are plotted. (**G**) sg*ROSA*- or sg*Huwe1*-RAW264.7-Dox-Cas9 cells were treated for 2 hr with LPS, followed by CHX chase in the continued presence of LPS. Protein lysates were harvested at the indicated time points (**h**). Endogenous TTP levels were measured by WB, quantified, plotted, and TTP half-life calculated. Data represent means and s.d.; n=3 biological replicates.

The online version of this article includes the following source data and figure supplement(s) for figure 2:

**Source data 1.** Blots corresponding to *Figure 2C* and to *Figure 2—figure supplement 1B*.

**Source data 2.** Blots corresponding to *Figure 2F*.

**Source data 3.** Blots corresponding to *Figure 2G*.

**Source data 4.** Cas9 functionality and leakiness evaluation.

**Source data 5.** Adopted gating strategy of FACS-based mCherry-TTP stability regulator screen.

**Source data 6.** MAGeCK analysis of D3 and D6 CRISPR screen in RAW-Dox-Cas9.

**Figure supplement 1.** Genome-wide CRISPR-Cas9 knockout screen identified the E3 ligase HUWE1 as a regulator of TTP stability.

**Figure supplement 1—source data 1.** Blots corresponding to *Figure 2—figure supplement 1F*.

**Figure supplement 1—source data 2.** Blots corresponding to *Figure 2—figure supplement 1G*.

**Figure supplement 1—source data 3.** Blots corresponding to *Figure 2—figure supplement 1H*.

candidates increased endogenous TTP and exogenous mCherry-TTP protein levels by western blot (*Figure 2—figure supplement 1B*), attesting to the validity and predictive quality of our screen.

In particular, HUWE1 was identified as a strong determinant of endogenous and exogenous TTP protein abundance by western blot (*Figure 2C* and *Figure 2—figure supplement 1B*; compare LPS-treated samples), and flow cytometry (*Figure 2D*), without affecting *Zfp36* mRNA levels (*Figure 2—figure supplement 1C*). Consistent with an increase in protein stability, inducible *Huwe1* knock-out significantly increased endogenous TTP protein levels at later time points post-LPS stimulation (*Figure 2E*). Given that after initial stabilization, LPS mediates TTP degradation (*Figure 1E*), we tested whether *Huwe1* knock-out affected TLR4 signaling. To this end, the effect of *Huwe1* loss on IκBα, which is degraded in a proteasome-dependent manner upon LPS stimulation (*Figure 2—figure supplement 1D*), was measured. *Huwe1* knock-out increased TTP protein levels (*Figure 2—figure supplement 1E*; top panel), but did neither affect LPS-induced degradation of mCherry-IκBα by flow cytometry (*Figure 2—figure supplement 1E*; bottom panel), nor endogenous IκBα by western blot (*Figure 2—figure supplement 1F*). This shows that the *Huwe1* knock-out does not affect cell signaling between TLR4 and IκBα, and this does not contribute to TTP stabilization in *Huwe1*-deficient cells.

Next, we determined whether the HUWE1-dependent control of TTP abundance in the RAW264.7 mouse macrophage cell line was conserved across species and cell types. To this end, *HUWE1* was targeted in the human colon carcinoma cell line RKO, which -unlike most myeloid cells- have low detectable levels of TTP in the absence of any stimulation (*Figure 2—figure supplement 1G*). Similar to the phenotype in RAW264.7 cells, *HUWE1* knock-out in RKO cells strongly increased TTP protein levels (*Figure 2—figure supplement 1G*), indicating that HUWE1 has a similar role in human, non-myeloid cells independent of the TLR4 axis. Moreover, targeting of *Huwe1* in mouse bone marrow derived macrophages (BMDMs), likewise strongly increased high and low molecular weight species of endogenous TTP (*Figure 2F*), indicating that the biological importance of *Huwe1* for TTP abundance is relevant in primary cells.

Lastly, we measured whether *Huwe1* ablation affected TTP protein half-life. To this end, *sgROSA* and *sgHuwe1* RAW264.7-Dox-Cas9 cells were continuously stimulated with LPS, chased in the

presence of translation inhibitor cycloheximide (CHX), analyzed by western blot, and single-step exponential decay curves plotted. Endogenous TTP was stabilized ≥5-fold in the absence of *Huwe1* (estimated half-life of ~20 hr), compared to sg*ROSA* cells in which TTP half-life was measured to be 3.7 hr (*Figure 2G*). In similar stability assessments with exogenously expressed TTP in the absence of LPS, *Huwe1* knock-out increased HA-TTP protein half-life by 83% from 35 min to 55 min (*Figure 2— figure supplement 1H*). Together, these data demonstrate that loss of *Huwe1* increases TTP protein half-life, and positioned HUWE1 as a strong, conserved regulator of TTP protein stability.

## Loss of *Huwe1* decreases the half-life of pro-inflammatory mRNAs controlled by TTP

TTP is essential for the degradation of transcripts with AU-rich elements in their 3'-UTR, encoding pro-inflammatory cytokines such as TNF and IL6. Phosphorylation of S52 and S178 stabilizes TTP, yet reduces its degradation of mRNAs. In contrast, the effect of phosphorylation on other sites has remained elusive. Therefore, we reasoned that increased TTP protein levels upon *Huwe1* ablation could (i) either result in increased intracellular TTP protein concentrations, and consequently diminished levels of transcripts encoding pro-inflammatory cytokines, or -as a consequence of increased TTP phosphorylation- (ii) decrease the bio-active pool of TTP, resulting in equal or increased mRNA levels in *Huwe1* knock-out cells.

To investigate whether increased TTP levels upon *Huwe1* loss are biologically relevant, we measured *Tnf* and *Il6* mRNA levels in *Huwe1*-targeted BMDMs and RAW264.7 cells. Consistent with the fact that non-stimulated cells have very low levels of TTP, *Huwe1* knock-out did not alter baseline *Tnf* (*Figure 3A*, and *Figure 3—figure supplement 1A*), or *Il6* (*Figure 3—figure supplement 1B/C*) mRNA levels.

LPS stimulation transcriptionally induces *Tnf* and *Il6*, and in parallel influences TTP protein stability. Loss of *Huwe1* resulted in significantly decreased concentrations of *Tnf* and *Il6* transcripts at 3–16 hr post-stimulation (*Figure 3A*, and *Figure 3—figure supplement 1A–C*), consistent with increased TTP protein levels in *Huwe1* knock-out cells (*Figure 2C*). In line with TTP-dependent post-transcriptional effects, *Huwe1*-loss only altered mature *Tnf* mRNA concentrations, whereas its pre-mRNA levels remained unaffected (*Figure 3B*), indicating that *Tnf* transcription was likely unaffected. Neither did any differences stem from altered macrophage differentiation from bone marrow, as no differences in F4/80 surface expression were measured between sg*ROSA* and sg*Huwe1* BMDMs (*Figure 3—figure supplement 1D*). Instead, Actinomycin D mRNA chase experiments indicated that the decreased levels of pro-inflammatory mRNAs are consistent with a 57% decrease in *Tnf* mRNA stability (*Figure 3C*). Moreover, targeting of *Zfp36* in *Huwe1*-deficient cells partially rescued *Tnf* mRNA concentrations (*Figure 3—figure supplement 1E*), indicating that the effects of *Huwe1* loss on *Tnf* mRNA levels are at least in part TTP-dependent. Lastly, measurements of intracellular cytokines by flow-cytometry showed that the decreased *Tnf* mRNA levels in *Huwe1* KO cells, were matched by significantly decreased intracellular TNF protein (*Figure 3—figure supplement 1F*).

Taken together, these data show that loss of *Huwe1* increases the bio-active pool of cellular TTP, resulting in enhanced turn-over of TTP target mRNAs encoding pro-inflammatory mediators.

## HUWE1 regulates TTP phosphorylation and its increase is responsible for increased TTP stability

Since HUWE1 is a ubiquitin E3 ligase and was identified as a regulator of TTP protein stability by genetic means, we reasoned that the effects from its ablation on TTP could be direct through complex formation and ubiquitination of TTP, or indirect by influencing the activity or abundance of proteins that regulate TTP. Neither co-IP, nor TurboID proximity labeling assays identified complex formation between TTP and HUWE1 in cells (*Figure 1—figure supplement 1I/J*).

This suggested that the effects of HUWE1 on TTP may be indirect, although direct ubiquitination of TTP by HUWE1 cannot be ruled out as their interaction may have been too transient to detect in our assays. Attempts to address direct TTP ubiquitination by HUWE1, or any of the other E3 ligases identified in the genetic screen (*Figure 2B* and *Figure 2—figure supplement 1A–B*; VHL, UBE3C, and the Cullin adapters Elongin B/C) were hindered by the inability to purify sufficient amounts of recombinant TTP protein.

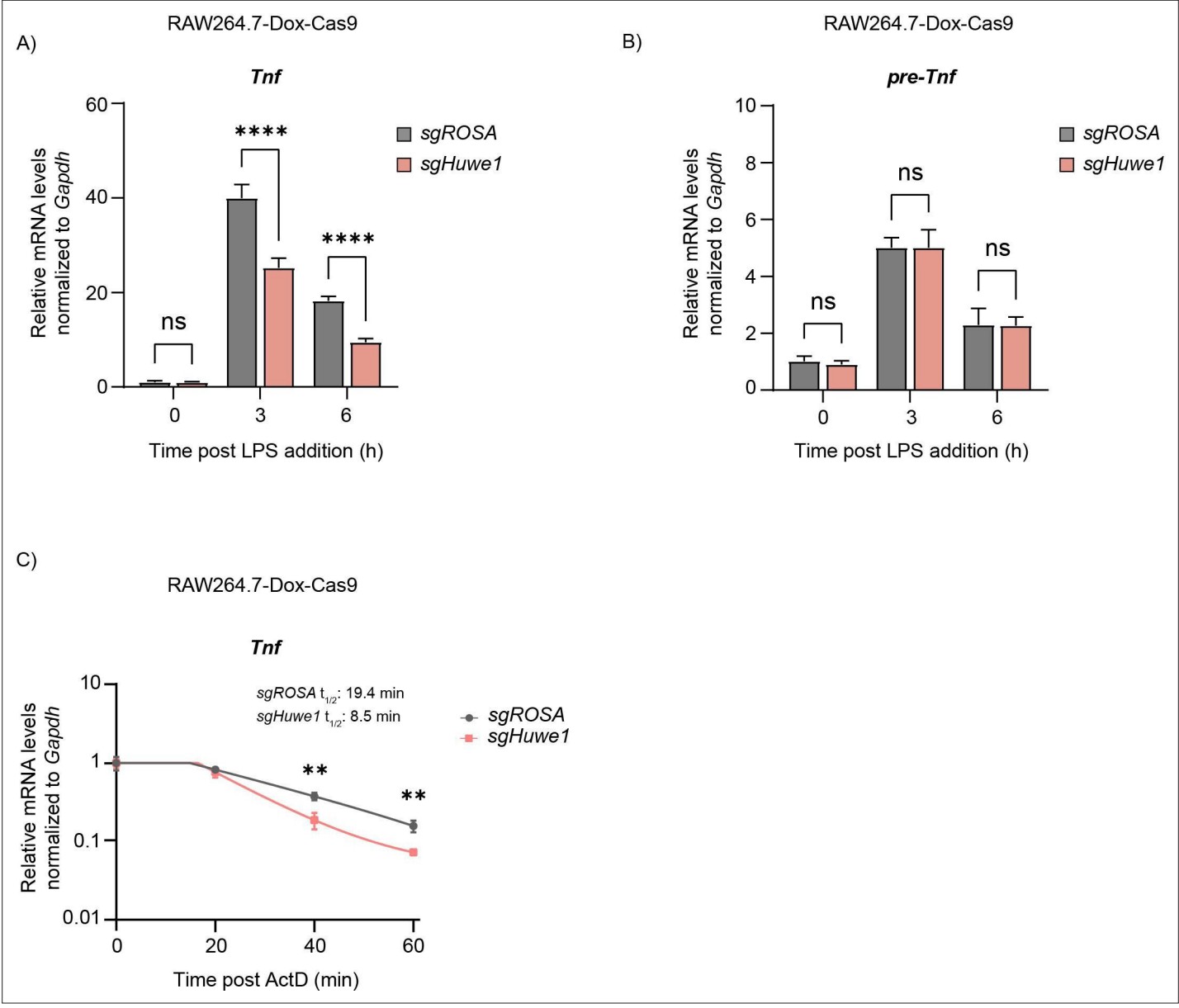

**Figure 3.** TTP mRNA targets are dysregulated upon *Huwe1* depletion. RAW264.7-Dox-Cas9 cells expressing sg*ROSA* or sg*Huwe1* were treated with Dox for 5 days to induce Cas9. Cells were incubated with LPS for the indicated time points (h). (**A**) Mature *Tnf* mRNA levels, and (**B**) *Tnf* pre-mRNA levels were measured by RT-qPCR and normalized to *Gapdh*. Data represent the mean and s.d.; n=3 biological replicates. ****p ≤0.0001. Two-way ANOVA was performed. (**C**) RAW264.7-Dox-Cas9 cells expressing sg*ROSA* or sg*Huwe1* were treated with Dox for 5 days to induce Cas9. Cells were incubated with LPS for 3 hr, after which Actinomycin D (ActD) was added for the indicated times (min), and *Tnf* mRNA levels were determined by RT-qPCR. Data represent the mean and s.d.; n=3 biological replicates. **p ≤0.01. Unpaired t-tests were performed for the 40 min and 60 min time points.

The online version of this article includes the following figure supplement(s) for figure 3:

**Figure supplement 1.** TTP mRNA targets are dysregulated upon *Huwe1* depletion.

Since TTP stability is regulated for an important part through phosphorylation by the stress kinase p38-MK2 axis and to a lesser extent ERK (**Brook et al., 2006**; **Deleault et al., 2008**; **Ronkina et al., 2019**), we set out to determine whether *Huwe1* ablation would alter TTP levels indirectly by affecting the cellular concentrations or activity of these kinases. Data from *Huwe1* knock-out cells indicated that the effect of *Huwe1* loss on TTP stability was predominantly at time points after the initial two hours of LPS stimulation (**Figure 2E**) during which TTP dephosphorylation of S52/S178 happens, resulting in its degradation (**Kratochvill et al., 2011**; **Sedlyarov et al., 2016**). Consistent with this finding, *Huwe1*

ablation did not significantly alter the total protein levels or change the early phosphorylation/activation kinetics of stress kinases p38, MK2, ERK, and JNK between 0 and 60 min post-LPS treatment (*Figure 4—figure supplement 1A–D*).

In contrast, ablation of *Huwe1* strongly increased endogenous TTP levels upon its induction by LPS at all measured later time points from 2 to 16 hr post-stimulation (*Figures 2E and 4A*). In the same lysates, total and activated/phosphorylated levels of p38, MK2, ERK, and JNK were determined.

The total levels of all four kinases varied slightly between the different time points post-LPS stimulation, yet these differences were independent of the targeted locus (*ROSA* or *Huwe1*). In contrast, *Huwe1*-targeting consistently increased the activated phosphorylated forms of all kinases at 2 hr post-stimulation (*Figure 4A–D*). While activated p38 and ERK levels in *Huwe1*-targeted cells were comparable to the *ROSA*-targeted control, or even lower, at later time points (*Figure 4A–C*), MK2 and JNK activation was increased for prolonged times, up to 6 hr post-stimulation (*Figure 4B–D*). Importantly, in the absence of LPS stimulation, *Huwe1* knock-out did not affect, or even decreased, baseline phosphorylated levels of all kinases (*Figure 4A–D*). Moreover, *Huwe1* ablation in the presence of LPS did not induce unrelated stress responses, such as p53 activation (*Figure 4—figure supplement 1E*), indicating that loss of *Huwe1* does not induce a general stress response in the cell, but is specific for pro-inflammatory cellular conditions mediated by LPS. Together, these data indicated that in the absence of *Huwe1*, multiple stress kinases may be activated more, and for prolonged times.

Based on the increased levels of phosphorylated TTP (*Figure 2C*) and stress kinase activation in *Huwe1* knock-out cells, we hypothesized that increased phosphorylation of TTP by some or all of the four deregulated kinases could be responsible for the elevated TTP protein stability. In particular, p38 and its downstream target MK2 were prime candidates, given their importance in LPS-induced TTP stabilization through phosphorylation of S52 and S178.

We reasoned that if HUWE1-dependent stability effects on TTP occured through altering TTP phosphorylation, that either preventing TTP phosphorylation by kinase inhibition, or saturating TTP phosphorylation by phosphatase inhibition would negate TTP stabilization in *Huwe1* KO cells. To investigate whether the increased activity of the four individual kinases in *Huwe1*-targeted cells was causative for the increased TTP stability, a mixed genetic/inhibitor epistasis experiment was performed. To this end, endogenous TTP levels were assessed by intra-cellular staining in *ROSA*- or *Huwe1*-targeted cells, which were additionally treated with individual inhibitors of p38, MK2, ERK, JNK, or a combination of all four inhibitors (*Figure 4E*).

Consistent with our other results, *Huwe1* knock-out increased TTP protein levels in 6 hr LPS-treated DMSO control cells (*Figure 4E*; sample set 3). In line with previous reports that the p38-MK2 axis is an important determinant of TTP stability, treatment of sg*ROSA*-targeted cells with p38 or MK2 inhibitors significantly decreased TTP levels (*Figure 4E*; sample sets 4 and 5). However, simultaneous *Huwe1* knock-out still elevated TTP levels in the presence of these individual inhibitors, indicating that either HUWE1 does not affect TTP through the p38-MK2 axis, or that there are compensatory mechanisms affecting TTP stability in the absence of p38-MK2 kinase activity.

Consistent with previous findings of a minor effect of ERK activity, and no effect of JNK on TTP stabilization (*Deleault et al., 2008*), ERK or JNK inhibition did not influence TTP levels (*Figure 4E*; sample sets 6 and 7). Moreover, the levels of TTP in the presence of either ERK or JNK inhibitors were still increased upon *Huwe1* knock-out, indicating that the activity of neither of these individual kinases alone is required for the HUWE1-dependent effect on TTP.

We hypothesized that the deregulated increase in activity of multiple of these four stress kinases could in a partially functionally compensatory manner contribute to elevated TTP phosphorylation (*Figure 4A–D*) and stability. Indeed, inhibition of all four kinases (4i) simultaneously rendered TTP highly unstable as expected, yet in contrast to the single kinase inhibitors, this was no longer affected by *Huwe1* knock-out (*Figure 4E*; sample set 8, and *Figure 2F*). From these data, we concluded that the HUWE1 effect on TTP stability is dependent on the activity of multiple stress kinases. Together, these results indicate that HUWE1 is important for curtailing TTP phosphorylation, thereby indirectly influencing TTP stability. We reasoned that the increased TTP phosphorylation in *Huwe1* knock-out cells could stem from either increased phosphorylation by the stress kinases, and/or decreased dephosphorylation.

Since MK2/p38, ERK, and JNK are activated/phosphorylated through independent cellular pathways, yet inactivated/dephosphorylated by the same phosphatases as TTP itself (PP1/2) (*Kruse et al.,*

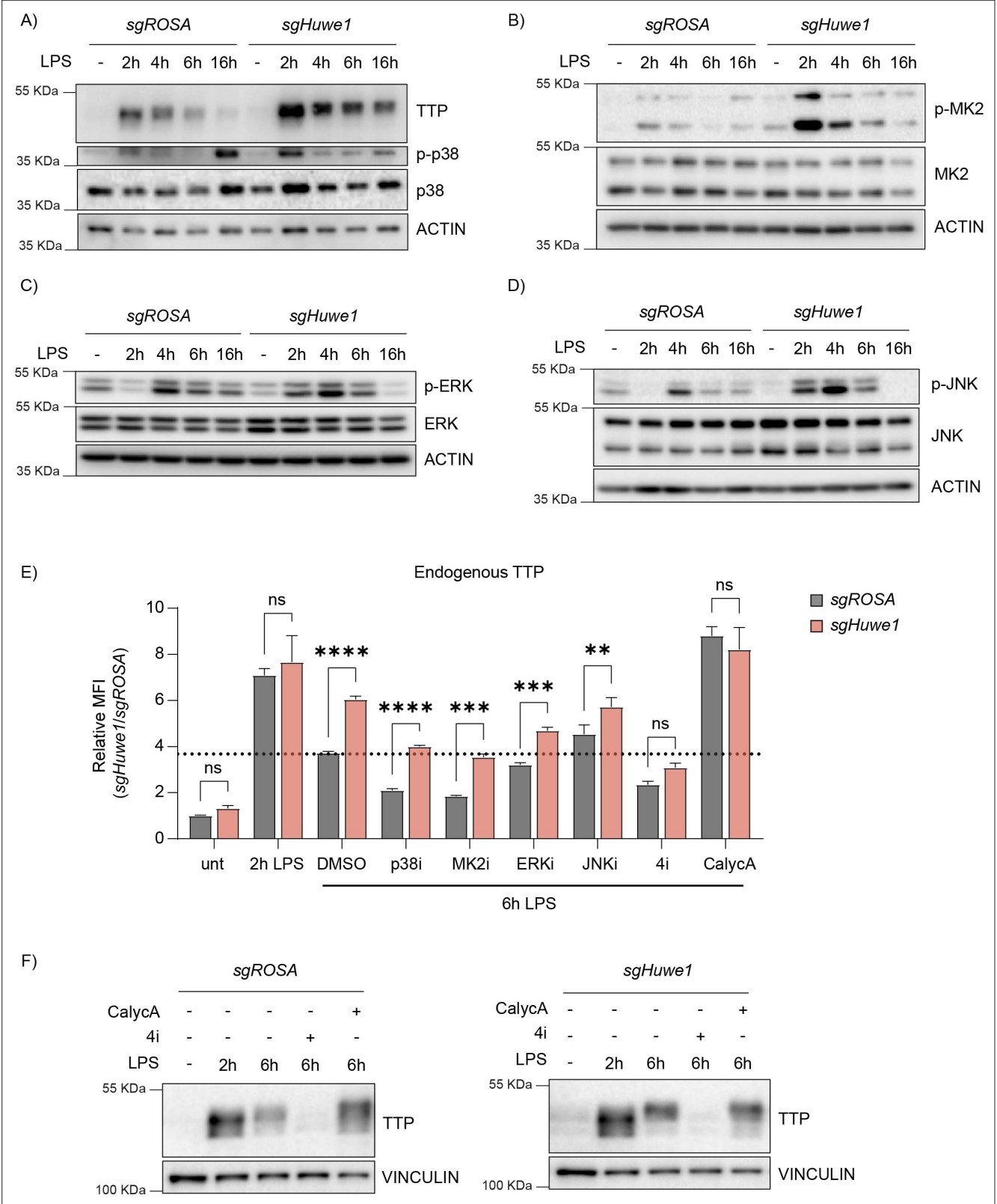

**Figure 4.** HUWE1 regulates TTP phosphorylation status, and thereby TTP stability. (**A–D**) RAW264.7-Dox-Cas9 cells expressing sg*ROSA* or sg*Huwe1* were treated with Dox for 5 days to induce Cas9. Cells were incubated with LPS for the indicated time points (h). Phosphorylation of (**A**) p38, (**B**) MK2, (**C**) ERK, and (**D**) JNK was determined by western blot. (**E**) sg*ROSA*- or sg*Huwe1*-RAW264.7-Dox-Cas9 cells were treated with LPS or left untreated. After 2 h of LPS treatment, cells were incubated with p38i, MK2i, ERKi, JNKi, or PP1/2 inhibitor Calyculin A (CalycA). TTP levels were analyzed by flow cytometry

*Figure 4 continued on next page*

*Figure 4 continued*

and normalized MFI plotted. Data represent the mean and s.d.; n=3 biological replicates. **p≤0.01; ***p≤0.001; ****p≤0.0001. Two-way ANOVA was performed. Dotted horizontal line indicates TTP abundance in the DMSO control at 6 hr post-LPS treatment. (**F**) sg*ROSA*- or sg*Huwe1*-RAW264.7-Dox-Cas9 cells were treated with LPS for the indicated times. During the last 4 hr of LPS stimulation, the indicated inhibitors were added, after which endogenous TTP levels were analyzed by WB.

The online version of this article includes the following source data and figure supplement(s) for figure 4:

**Source data 1.** Blots corresponding to *Figure 4A*.

**Source data 2.** Blots corresponding to *Figure 4B*.

**Source data 3.** Blots corresponding to *Figure 4C*.

**Source data 4.** Blots corresponding to *Figure 4D*.

**Source data 5.** Blots corresponding to *Figure 4F*.

**Figure supplement 1.** HUWE1 regulates TTP phosphorylation status, and thereby its TTP stability.

**Figure supplement 1—source data 1.** Blots corresponding to *Figure 4—figure supplement 1A*.

**Figure supplement 1—source data 2.** Blots corresponding to *Figure 4—figure supplement 1B*.

**Figure supplement 1—source data 3.** Blots corresponding to *Figure 4—figure supplement 1C*.

**Figure supplement 1—source data 4.** Blots corresponding to *Figure 4—figure supplement 1D*.

**Figure supplement 1—source data 5.** Blots corresponding to *Figure 4—figure supplement 1E*.

---

*2020*; *Nguyen and Shiozaki, 1999*; *Takekawa et al., 1998*; *Takekawa et al., 2000*; *Warmka et al., 2001*), we reasoned that it was most likely that HUWE1 may be important to regulate PP1/2 activity or its cellular concentrations. Therefore, we hypothesized that decreased PP1/2 output in *Huwe1* knock-out cells could prolong TTP phosphorylation by: (**i**) diminishing direct TTP dephosphorylation by PP1/2, and (**ii**) indirectly prolonging stress kinases activation as a consequence of their diminished dephosphorylation by PP1/2.

To test this hypothesis, *sgROSA* or *sgHuwe1* cells were treated with LPS for 6 hr, and from 2 hr onward, co-incubated with PP1/2 inhibitor Calyculin A (*Figure 4E–F*). As expected, preventing dephosphorylation by this inhibitor stabilized TTP, and prevented TTP degradation by 6 hr of LPS treatment (*Figure 4E*; compare sg*ROSA* 6 hr LPS with sg*ROSA* 6 hr LPS +CalycA; sample set 9, and *Figure 2F*). In contrast to sg*Huwe1* samples treated for 6 hr with LPS (in which TTP protein levels were increased), *Huwe1* knock-out no longer increased TTP protein concentrations in the presence of Calyculin A (*Figure 4E*; compare sg*Huwe1* 6 hr LPS with sg*Huwe1* 6 hr LPS +CalycA, and *Figure 2F*).

From these results, we conclude that in the absence of HUWE1, decreased cellular output of PP1/2 may prolong stress kinase activation. Increased kinase activity and decreased dephosposphorylation of TTP by PP1/2 consequently increases TTP phosphorylation, thereby stabilizing it.

## HUWE1 controls only a small fraction of proteasome targets, and regulates the abundance of TTP paralog ZFP36L1

HUWE1 has been shown to associate with proteasomes (*Besche et al., 2009*), the biological significance of which has remained elusive. We reasoned that HUWE1 might be important for proteasome activity, and its ablation could cause a general impaired degradation of proteasome targets such as TTP. To investigate whether this was the case, we compared the proteomes of LPS-stimulated RAW264.7-Dox-Cas9 cells in which we targeted either *Huwe1* or proteasome core particle component *Psmb7* by label-free mass-spectrometry.

As expected, *Psmb7* targeting altered the abundance of a large number of proteins, many of which are known targets of proteasomal degradation (*Figure 5—figure supplement 1A* and *Figure 5—source data 2*). In contrast, *Huwe1* ablation significantly changed the concentrations of only a select number of proteins (*Figure 5A* and *Figure 5—figure supplement 1B*). In line with expectations of an E3 ligase, HUWE1 targets showed a trend of also being increased in *Psmb7* knock-out cells, and *vice versa* (*Figure 5A* and *Figure 5—figure supplement 1A/B*). However, there was no clear correlation between the most affected proteins in the two genotypes, indicating that HUWE1 is likely not essential for proteasome function in cells, and that the increase of TTP in *Huwe1* knock-out cells is unlikely to have resulted from diminished overall proteasome activity. Among the differentially regulated

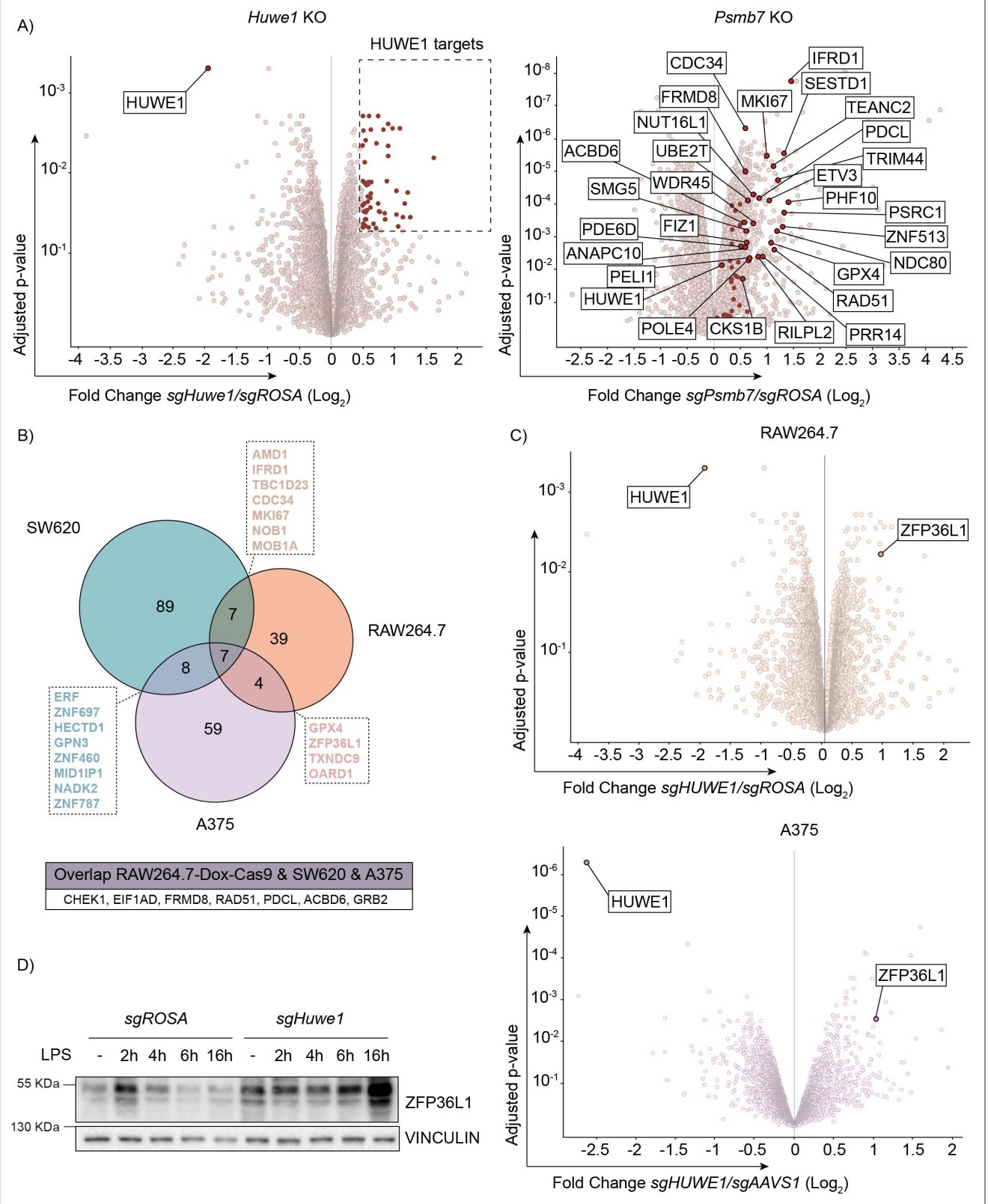

**Figure 5.** TTP family member ZFP36L1 abundance is increased upon *HUWE1* knockout in human and mouse cells. (**A**) RAW264.7-Dox-Cas9 expressing sg*ROSA*, sg*Huwe1* or sg*Psmb7* were treated with Dox for 3 days to induce Cas9. Proteome changes were assessed by quantitative mass spectrometry. Proteins classified as HUWE1 targets are highlighted in red. Shared HUWE1 and proteasome targets are labelled in the *Psmb7* knock-out volcano plot. (adjusted p-value ≤0.05 and Fold Change (Log$_2$) ≥0.5; n=3 biological replicates). (**B**) Venn diagram showing the overlap between proteome changes of

*Figure 5 continued on next page*

Figure 5 continued

*Huwe1*-targeted RAW264.7, A375, and SW620 cell lines. Shared targets are listed (adjusted p-value ≤0.05 and Fold Change (Log$_2$) ≥0.5; n=3 biological replicates). (**C**) Volcano plots representing proteome changes of *Huwe1*- and *AAVS1/ROSA*-targeted A375 human melanoma cells and RAW264.7-Dox-Cas9 cells (adjusted p-value ≤0.05 and Fold Change (Log$_2$) ≥0.5; n=3 biological replicates). The shared HUWE1 target ZFP36L1 is highlighted. (**D**) sg*ROSA* or sg*Huwe1* knockout RAW264.7-Dox-Cas9 cells were treated with Dox for 5 days to induce Cas9, followed by LPS treatment for the indicated times (h). Endogenous ZFP36L1 protein levels were determined by western blot.

The online version of this article includes the following source data and figure supplement(s) for figure 5:

**Source data 1.** Blots corresponding to *Figure 5D*.

**Source data 2.** Quantitative proteomics to systematically assess protein changes after HUWE1 knockout, in RAW264.7-Dox-Cas9, A375, and SW620 cell lines.

**Figure supplement 1.** Abundance of the TTP family member ZFP36L1 is increased upon *HUWE1* knockout in human and mouse.

proteins were factors previously identified as HUWE1 targets (*Cassidy et al., 2020*; *Thompson et al., 2014*; *Xu et al., 2016*), including GRB2, CHEK1, and CDC34 (*Figure 5—figure supplement 1C*).

Our data so far indicated that HUWE1 is important for proper regulation of TTP phosphorylation, and that in its absence the equilibrium shifted to a hyper-phosphorylated state (*Figure 4*), dependent on a decrease in phosphatase activity, and an increase in stress kinase activity, without major effects on their protein levels.

To further assess in an unbiased manner whether *Huwe1* deficiency would affect MAPK or PP1/2 protein levels, we extended our proteome mass-spectrometry for two additional human cell lines (A375 and SW620). Consistent with our previous findings (*Figure 4*), *Huwe1* ablation did not substantially or consistently affect the protein levels of detected MAPK or PP1/2 subunits in the three cell lines (*Figure 5—figure supplement 1D*/E). In line with the data presented above (*Figure 4*), this suggests that the hyper-phosphorylated TTP state in *Huwe1*-targeted cells does not result from changes in MAPK or PP1/A protein levels.

Previous studies have indicated that HUWE1 can target broader classes of cellular substrates (*Grabarczyk et al., 2021*; *Hunkeler et al., 2021*), but that the targeted proteins may be cell type specific to some degree. Analysis of proteome changes in the three different cell lines identified seven proteins that were consistently increased in all *Huwe1*-targeted cell lines (*Figure 5B*). Moreover, the protein concentration of other proteins was changed in only two of the three cell lines, whereas it was not detected in the third (*Figure 5B*).

We reasoned that any of these common deregulated proteins in *Huwe1* knock-out cells could contribute to the TTP hyper-phosphorylation/stabilization phenotype. However, analysis of overlap between factors that regulate TTP abundance identified in the genetic screen (*Figure 2B* and *Figure 2—figure supplement 1A*), and proteins deregulated by *Huwe1*-ablation did not identify any overlap, suggesting that these *Huwe1*-regulated proteins are unlikely to drive the effect on TTP protein stability.

Importantly, proteome measurements by mass-spectrometry are limited to detection of only reasonably abundant proteins. Even after LPS stimulation, no TTP peptides were identified in any of the three analyzed cell lines (*Figure 5B*), indicating that its absolute intra-cellular concentrations in these cells are too low to be detected by this method. In contrast, peptides of its paralog ZFP36L1 were readily identified in RAW264.7 and A375 cells and among the most increased proteins identified in *Huwe1*-targeted cells (*Figure 5C*). In line with this observation, independent western blot analysis of ZFP36L1 in cell lysates from *Huwe1*-deficient RAW264.7 cells showed that ZFP36L1 protein levels were increased in *Huwe1* knock-out cells (*Figure 5D*).

Collectively, these findings indicate that *Huwe1* ablation does not alter MAPK or PP1/2 protein levels, but that rather their differential activation alters TTP phosphorylation and stability. Moreover, ZFP36L1 abundance is regulated by HUWE1, akin to its closest related family member -TTP-, indicating that they could be regulated by HUWE1 in a conserved manner.

## Residues in the TTP 234-278 region are important for its stability

Lysines are exclusively located in the TTP zinc finger domain, and our data indicate that this is the site of poly-ubiquitination (*Figure 1—figure supplement 1C–G*). In line with this notion, upon mutation of the five lysine residues in its zinc finger domain (KtoR), TTP accumulated as a stable, phosphorylated

species (*Figure 6A–B* and *Figure 6—figure supplement 1A–B*). Moreover, this mutant was no longer affected by *HUWE1* loss, indicating that the effects of HUWE1 on TTP stability are dependent on ubiquitination in the zinc finger domain.

We reasoned that the E3 ligase ubiquitinating TTP at that site could bind TTP in its folded zinc finger domain, or that the mRNA-engaged TTP pool could be the predominant HUWE1-dependent target. Therefore, we addressed whether a TTP mutant with a disrupted zinc finger domain (C116R, C139R; *Lai et al., 2018*; *Ming et al., 2001*) would be stabilized. Zinc finger domain disruption did not accumulate at higher steady-state levels than its wtTTP counter-part, and was still increased upon *HUWE1* loss (*Figure 6B* and *Figure 6—figure supplement 1B*), demonstrating that neither recognition of the folded zinc finger domain structure by an E3 ligase, nor TTP functionality are required for the HUWE1 effects.

Our data support a role of HUWE1 in determining TTP phosphorylation, and thereby its stability. Therefore, we next analyzed whether phosphorylation of the two best-characterized TTP residues in this context (S52, S178) are important for HUWE1 effects. As for the zinc finger domain mutant, a S52A/S178A TTP mutant was still stabilized by *HUWE1* loss (*Figure 6B* and *Figure 6—figure supplement 1B*/C). This indicated that while phosphorylation of these residues importantly controls TTP stability, HUWE1 effects are likely independent of these phospho-residues.

Together, these data from full-length TTP point mutants suggest that an unknown E3 ligase likely binds TTP outside of its zinc finger domain, but ubiquitinates it on lysines inside the zinc finger domain. Moreover, we concluded that HUWE1 regulation of TTP stability and phosphorylation is independent of the MK2-stabilized S52/S178 residues. This is consistent with the finding that TTP levels were still increased in *Huwe1* knock-out cells treated with inhibitors of the p38 and MK2 kinases phosphorylating these two sites (*Figure 4E*; subsets 4–5). Lastly, these data indicate that TTP does not require an intact zinc finger domain for stability, suggesting that TTP engagement with target mRNAs is likely not a prerequisite for HUWE1-dependent stability regulation.

Next, we set out to determine which part of TTP regulates its HUWE1-dependent phosphorylation and stability. To this end, progressive N- and C-terminal TTP deletion mutants (*Figure 6A*) were analyzed in cells for their steady-state concentrations, phosphorylation, and sensitivity to *HUWE1* ablation. Since TTP is predicted to be mostly disordered outside of its zinc finger domain (https://alphafold.ebi.ac.uk/entry/P22893), we reasoned that the effects of the truncations on overall protein structure would be limited.

N-terminal deletions did neither affect TTP protein levels, its phosphorylation, nor the effect of *HUWE1* loss (*Figure 6C* and *Figure 6—figure supplement 1D–E*; deletions ΔN1-4), indicating that HUWE1 does not influence TTP stability through residues N-terminal of the zinc finger domain. Likewise, the two most C-terminal deletions did also not affect TTP stability (*Figure 6D* and *Figure 6—figure supplement 1F*; ΔC1-2). In contrast, further truncation of the C-terminus rendered mutant ΔC3 (259–278 region) less sensitive to *HUWE1* loss, yet retained its heterogeneous size distribution for phosphorylated species. Further deletion of the 234–258 region in the ΔC4 mutant strongly stabilized TTP at a homogeneously phosphorylated size, and rendered it insensitive to *HUWE1* knock-out (*Figure 6D* and *Figure 6—figure supplement 1F*). Likewise, the ΔC5 and ΔC6 mutants were insensitive to *HUWE1* knock-out, but accumulated as unphosphorylated TTP species. Together, these data indicate that the 234–278 region (*Figure 6D* and *Figure 6—figure supplement 1F*) is important for HUWE1-dependent regulation of TTP stability, and its phosphorylation status.

Since the TTP-ΔC3 mutant was stabilized in a HUWE1-insensitive manner (*Figure 6D* and *Figure 6—figure supplement 1F*), we reasoned that this region (259-278) could be important for proteasomal targeting (e.g. an E3 ligase binding site). In contrast, the TTP-ΔC4 mutant accumulated as a lower MW homogenously phosphorylated TTP species (*Figure 6D* and *Figure 6—figure supplement 1F*), suggesting that the 234–258 region regulates TTP stability by affecting its phosphorylation status (e.g. a phosphatase binding site).

To test these possibilities, we analyzed mutants in which either only the 259–278 or 234–258 regions were deleted, while retaining the rest of the protein (*Figure 6A*). Consistent with the data from *Figure 6D*, a TTP mutant only lacking the 259–278 region was strongly stabilized, accumulated predominantly as a relatively homogeneous phosphorylated species, and was insensitive to *HUWE1* knock-out (*Figure 6E* and *Figure 6—figure supplement 1G*). Moreover, deletion of the 234–258

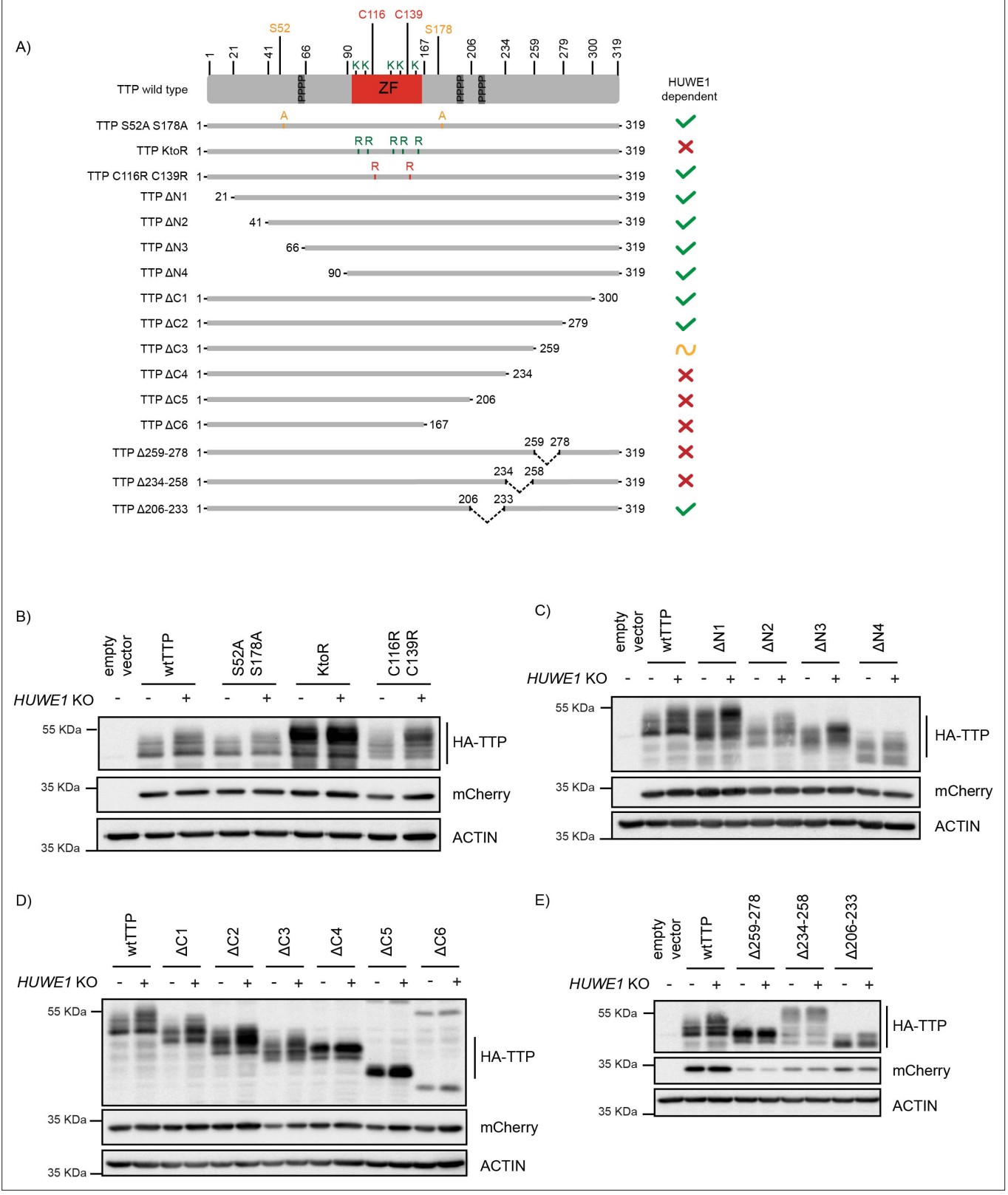

**Figure 6.** Residues in the TTP 234–278 region are important for its stability. (**A**) Schematic representation of 3xHA-TTP mutants. Colors denote amino acid substitutions. ZF indicates the zinc finger domain, and the three tetraprolin motifs are presented as dark grey boxes. (**B–E**) sg*AAVS1-* and sg*HUWE1*-depleted HEK-293T-Cas9 cells were transfected with the indicated mutants, and 3xHA-TTP stability was determined by western blot. mCherry is expressed as a stable internal control through a P2A site.

*Figure 6 continued on next page*

*Figure 6 continued*

The online version of this article includes the following source data and figure supplement(s) for figure 6:

**Source data 1.** Blots corresponding to *Figure 6B*.

**Source data 2.** Blots corresponding to *Figure 6C*.

**Source data 3.** Blots corresponding to *Figure 6D*.

**Source data 4.** Blots corresponding to *Figure 6E*.

**Figure supplement 1.** Residues in the TTP 234–278 region are important for its stability.

**Figure supplement 1—source data 1.** Blots corresponding to *Figure 6—figure supplement 1C*.

**Figure supplement 1—source data 2.** Blots corresponding to *Figure 6—figure supplement 1E*.

region resulted in TTP hyper-phosphorylation (*Figure 6E* and *Figure 6—figure supplement 1G*), consistent with the idea of it being important for phosphatase binding.

Together, these data indicate that the TTP ΔC3-specific region (259-278) is consistent with a possible binding site for an E3 ligase, whereas the ΔC4-specific region (234-258) is a likely interaction site of a phosphatase (*Figure 7A*). Importantly, the ΔC5- and ΔC6-mutants were stabilized, yet not hyperphosphorylated (*Figure 6B*). This suggests that the phosphorylated residues contributing to TTP stabilization in *HUWE1* knock-out cells are likely in the possible E3 ligase binding site in the TTP ΔC3-specific region (259-278) (*Figure 7A*).

In summary, we provide evidence for ubiquitin-dependent proteasomal degradation as a key regulatory mechanism for TTP protein abundance in cells. A genetic screen identified HUWE1 as a strong regulator of TTP proteasomal turn-over. In the absence of *Huwe1*, TTP is heavily phosphorylated and stabilized, which is dependent on multiple ubiquitination sites in the TTP zinc finger domain, and phosphorylation in the 259–278 region. We propose that this region in its unphosphorylated form is also a likely binding site for an E3 ligase directing TTP ubiquitination and degradation (*Figure 7A*). Moreover, the adjacent 234–258 region is consistent with an interaction site for the main TTP phosphatases (PP1/2) (*Figure 7A*).

We propose a model in which HUWE1 under physiological conditions curtails stress kinase activation, thereby limiting their stabilizing effects on TTP (*Figure 7B*). However, in the absence of *Huwe1*, the collective activity increase of these stress kinases results in TTP hyper-phosphorylation in the 259–278 region, increased TTP stability, and decreased pro-inflammatory output. Since we found that

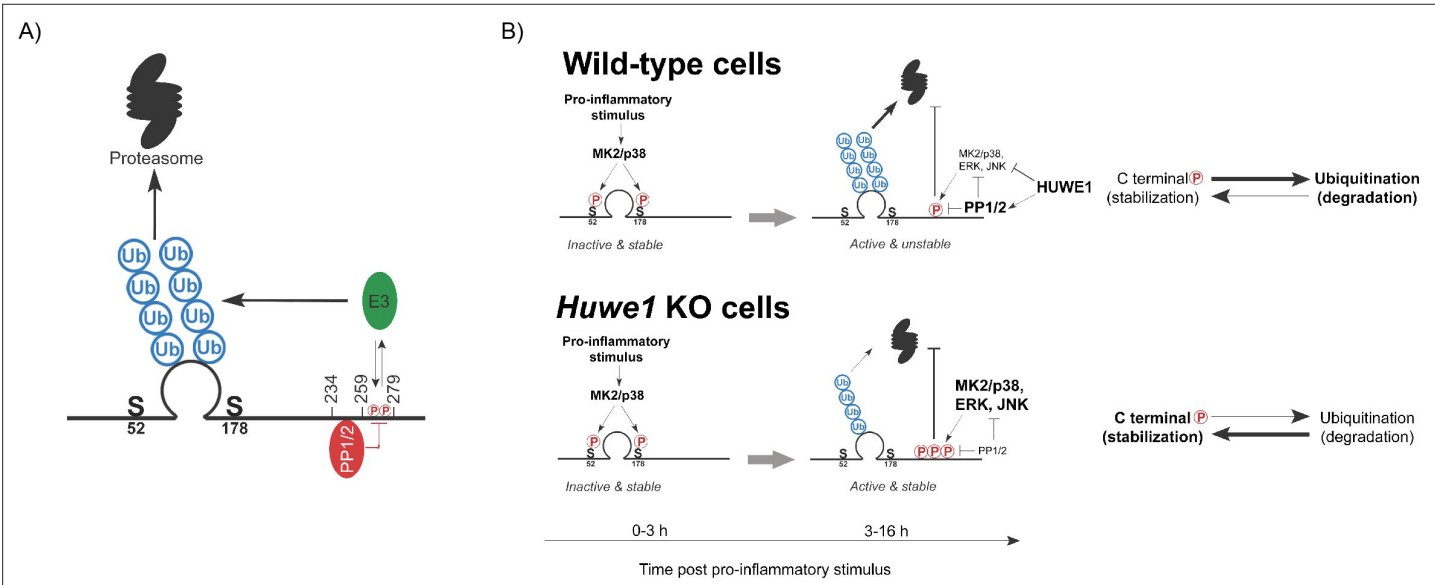

**Figure 7.** Model of HUWE1-dependent TTP regulation. (**A**) Model indicating the TTP regions in its C-terminus speculated to recruit PP1/2 and an unknown E3 ligase that ubiquitinates the zinc finger domain. (**B**) Model of TTP stability regulation through phosphorylation in wild-type cells and *Huwe1*-deficient cells.

TTP phosphorylation is inversely correlated with ubiquitination and degradation, we speculate that phosphorylation in this region could prevent E3 ligase binding (*Figure 7A*).

## Discussion

Previous studies have predicted that TTP is disordered outside of its zinc finger domain, and showed that these unstructured regions contribute to its rapid proteasomal turn-over (*Ngoc et al., 2014*; *Ross et al., 2015*). Protein disorder is often associated with proteasomal turn-over, as these regions often contain degrons, accessible ubiquitination sites, or provide an initiation side for threading into the proteasome and initiating unfolding and translocation into the catalytic chamber (*Aufderheide et al., 2015*; *van der Lee et al., 2014*). However, previous work did not identify TTP poly-ubiquitination, and showed that incubation of TTP with purified 20S proteasomes -which lack the Ub-receptor containing 19 S regulatory particle-, were sufficient to degrade TTP (*Ngoc et al., 2014*).

Work from Ngoc et al. showed that fusion to GFP of either the N-terminal TTP part, or the TTP C-terminal part (aa 214–436), destabilized GFP in cells (*Ngoc et al., 2014*). Thus, the GFP destabilization was seemingly indiscriminate, and possibly caused by the disordered nature of the fusion construct per se. Since the C-terminal TTP part fused to GFP included aa 214–436, we cannot rule out that part of this effect was HUWE1-dependent. However, the discrepancy with our finding that the TTP N-terminus does not contribute to HUWE1-dependent TTP regulation, may suggest that the GFP fusions by Ngoc et al. were destabilized by more general protein principles, rather than HUWE1-specific effects.

It has been reported that oxidized, unfolded proteins could be directly degraded by 20S proteasomes (*Davies, 2001*; *Inai and Nishikimi, 2002*). However, the prevailing notion is that association of a regulatory particle is critical to open access to the catalytic chamber (*Coux et al., 1996*; *Davies, 2001*; *Driscoll and Goldberg, 1990*; *Eytan et al., 1989*), and efficient substrate degradation in cells. Here, we demonstrate robust poly-ubiquitination of TTP in denaturing RAW264.7 lysates, indicating that these poly-ubiquitin chains are covalently attached to TTP, and do not interact through a putative ubiquitin-interaction domain. Moreover, we show that mutation of lysines in the zinc finger domain stabilized TTP, and that an inhibitor preventing de novo ubiquitination in cells, stabilized TTP. These data demonstrate that poly-ubiquitination of TTP is essential for its degradation, which is likely mediated by 26S proteasomes.

In contrast to other published work (*Ngoc et al., 2014*), non-degrative TRAF2-driven K63-linked poly-ubiquitination of TTP has been previously reported to mediate the balance between NFκB and JNK-dependent signaling using transfected HEK-293T cells (*Schichl et al., 2011*). Consistent with TTP being rapidly proteasomally turned-over, we readily detected K48-linked poly-ubiquitination of TTP in the same cell system, yet failed to detect substantial K63-linked poly-ubiquitination. This dissimilarity could stem from differences in expression levels, lack of co-expression of TRAF2, or insufficicient sensitivity of the K63-specific antibody in our assays. As TNF and TLR activation activate NFκB and stress kinases -including JNK- in parallel, it will be of interest to further dissect in future studies the interplay between HUWE1-dependent phospo-regulation of TTP, and its K63-ubiquitination in stimulated myeloid cells.

Most proteins with disordered regions will eventually be degraded in in vitro reactions containing high concentrations of 20S proteasomes (*Liu et al., 2003*; *Lu et al., 2015*). This could explain the previous finding of ubiquitin-independent TTP degradation in vitro (*Ngoc et al., 2014*). Future comparisons of TTP degradation kinetics of ubiquitinated and non-ubiquitinated forms in the presence of 26S proteasomes will be important to further address this issue.

Several E3 ligases to putatively poly-ubiquitinate TTP were identified in our genetic screen (*Figure 2*): HUWE1, VHL, UBE3C, and the Cullin adapters Elongin B/C. *Huwe1* ablation most robustly stabilized TTP, based on which we hypothesized that HUWE1 may directly poly-ubiquitinate TTP. However, multiple independent techniques, including TurboID proximity labeling and co-IPs, failed to identify an interaction between TTP and HUWE1, which suggested that it may instead indirectly influence TTP stability.

At this point, we cannot rule out that HUWE1 directly poly-ubiquitinates TTP, resulting in its proteasomal degradation. Alternatively, one or more of the other identified E3 ligases could contribute to direct TTP ubiquitination. If indeed the other E3 ligases contribute to TTP ubiquitination, the fact that their knock-out phenotype is substantially less than that of *Huwe1* loss (*Figure 2—figure supplement*

*1B*) may suggest that multiple of them could be functionally redundant. Irrespective of potential direct TTP ubiquitination by HUWE1, multiple lines of evidence point towards a strong contribution of HUWE1-dependent differential TTP phosphorylation as an indirect means to control TTP stability and functional activity. This is consistent with our finding that TTP phosphorylation and ubiquitination appear to be inversely correlated, as the TTP KtoR mutant accumulates as a hyper-phosphorylated species (*Figure 6B*).

Published data support a model in which TTP upon translation is initially phosphorylated by the p38/MK2 kinase axis on residues S52 and S178 (mTTP numbering), resulting in its stabilization, yet repressing TTP function (*Deleault et al., 2008*; *Hitti et al., 2006*; *Kratochvill et al., 2011*; *Ross et al., 2015*). At later stages, diminishing p38/MK2 kinase activity is thought to shift the equilibrium to dephosphorylation of TTP at these residues, rendering it active, but unstable. Under these conditions, TTP is rapidly turned over by the proteasome (*Deleault et al., 2008*; *Ross et al., 2015*).

Consistent with these data, we also found that an S52A/S178A mutant is unstable in the absence and presence of LPS stimulation (*Figure 6B*, and *Figure 6—figure supplement 1C*). Importantly, this mutant was still stabilized in the absence of *Huwe1*, indicating that the stabilizing effects of phosphorylation on S52/S178 are independent of HUWE1. Moreover, it indicates that HUWE1-dependent effects on TTP stability target other sites in TTP. Thus, S52/S178 phosphorylation seems predominantly relevant for TTP stabilization at the early (2–3 hr) time points post-LPS-stimulation (*Ross et al., 2015*), whereas HUWE1-dependent effects occur later (between 3–16 hr) post-LPS stimulation. In contrast to S52/178, TTP phosphorylation in its C-terminal 259–278 region and its associated stabilization in the absence of *Huwe1*, is paralleled by decreased concentrations of known TTP mRNA targets, which suggests that phosphorylation in this region may not inhibit TTP functional output.

To uncouple LPS-induced *Zfp36* transcription from PTMs influencing TTP protein stability, we complemented experiments using endogenous TTP with exogenously expressed counterparts. In the absence of LPS stimulation, *Huwe1* loss did increase exogenous TTP, albeit rather mildly (*Figure 2C* and *Figure 2—figure supplement 1B*). In contrast, upon LPS stimulation and stress kinase activation, the effect of *Huwe1* knock-out was much stronger, resulting in strong TTP protein accumulation, which included a substantial fraction of phosphorylated forms (*Figure 2C and F*, and *Figure 2—figure supplement 1B*). These results suggest that there may be low baseline levels of activated stress kinases in the cell that affect TTP stability in the absence of LPS. However, in contrast to these mild phenotypes, the predominant effects of *Huwe1* loss on TTP protein stability and phosphorylation occur after LPS stimulation, in line with the notion that HUWE1 regulates TTP stability through influencing stress kinase-dependent TTP phosphorylation (*Figures 5 and 6*).

*Huwe1* loss increased the activity of multiple stress kinases (*Figure 4A–D*) without affecting their total protein levels. The previous findings that (**i**) the phosphorylation and activation of these kinases is controlled by PP2A (*Kruse et al., 2020*; *Nguyen and Shiozaki, 1999*; *Takekawa et al., 1998*), and (**ii**) the observation that combined inhibition of the stress kinases, or inhibition of PP1/2 activity with Calyculin A, rescued *Huwe1* knock-out effects on TTP (*Figure 4E–F*), suggest that HUWE1 may be important for PP1/2 activity. In line with this notion, decreased PP1/2 activity in *Huwe1* knock-out cells could affect TTP phosphorylation and stability by directly affecting its dephosphorylation, and in parallel maintain high stress kinase activity, which increases TTP phosphorylation even more.

This functional interaction between HUWE1 and the activity of PP1/2 and stress kinases has not been described previously, although it should be noted that HUWE1 has been reported to control the abundance and activity of other kinases and phosphatases (*Cassidy et al., 2020*; *Jang et al., 2014*; *Su et al., 2021*). Our findings broaden the understanding of how HUWE1 may indirectly influence numerous cellular proteins beyond direct recognition as ubiquitination substrates.

Taken together, our data support a model in which HUWE1 is important to maintain PP1/2 cellular output, and curtail stress kinase activation. This in turn limits phosphorylation in the TTP region spanning residues 259–278, allowing for recruitment of HUWE1 itself or another -yet unidentified- E3 ligase to that same region, subsequent poly-ubiquitination on lysine residues in the zinc finger domain, and ultimately proteasomal degradation. Phosphorylation in this region could prevent E3 ligase binding. Although the HUWE1-dependent phosphorylation effect appears to be dependent on the putative E3 ligase binding site (259-278), phosphatase recruitment to the 234–258 region in TTP seemingly controls dephosphorylation of most or all phospho-sites on TTP (*Figure 6C*). Since ZFP36L1 abundance is also strongly regulated by HUWE1, this suggests that its C-terminal region is orthologous to

TTP 234–278 and how it controls HUWE1-dependent degradation could be conserved across these family members.

# Materials and methods

## Vectors

The lentiviral mouse genome-wide sgRNA library (six sgRNAs/gene) has been described previously (*Michlits et al., 2020*). Lentiviral vectors driving the expression of a single sgRNA or a dual sgRNA from a U6 promoter, and either eBFP2 or iRFP from a PGK promoter have been described previously (*de Almeida et al., 2021*). Single sgRNA CDSs were cloned in pLentiCRISPRv2 (Addgene plasmid 52961) to perform stable knock-outs in HEK293T cells. A Dox-inducible Cas9 lentiviral vector was modified from LT3GEPIR (Addgene plasmid 111177): T3G-GFP-(miR-E)-PGK-Puro-IRES-rtTA3, in which the GFP-mirE cassette was replaced by Cas9-P2A-GFP from pLentiCRISPRv2. The TTP stability reporter (pLX-SFFV-mCherry-TTP-P2A-BFP) was constructed by cloning the open-reading frame (ORF) of murine TTP into a modified pLX303 vector (Addgene plasmid 25897). Lentiviral N-terminally HA-tagged-TTP deletions or point mutant variants were obtained by cloning the indicated variants of murine TTP ORF into a modified pLX303 vector. For this purpose, cDNAs encoding mTTP mutants were purchased from Twist Bioscience. All 3xHA-TTP constructs co-expressed mCherry through a P2A site to monitor protein expression and protein stability. All plasmids and sgRNAs used in this study are listed in *Table 1* and *Table 2*.

## Cell culture and reagents

All experiments in this study have been reproduced at least twice in independent experiments. Cell lines were tested negative for mycoplasma contamination. All cell lines used in this study and their applications are listed in *Table 3*. None of the used cell lines are on the current list of commonly misidentified cell lines (v12). Parental cell lines were obtained from ATCC: A375 (CRL-1619), RKO (CRL-2577), SW620 (CCL-227), and authenticated by short tandem repeat analysis. These cell lines were used to generate dox-inducible Cas9 derivatives as indicated in *Table 3*. RAW264.7 Dox-Cas9 cells were generated by transducing RAW264.7 cells with pRRL-TRE3G-Cas9-P2A-GFP-PGK-IRES-rtTA3 lentiviral vector. Cas9 expression was induced with 500 ng/ml of Docycycline hyclate (Dox, Sigma-Aldrich, D9891) and single cells were sorted by FACS into 96-well plates using a FACSAria III cell sorter (BD Biosciences) to obtain single-cell-derived clones. Cas9 function and leakiness of the TRE3G promoter in the absence of Dox was tested in competitive proliferation assays. For mCherry-TTP reporter cells, pLX303-SFFV-mCherry-TTP-P2A-BFP was transduced into RAW264.7-Dox-Cas9 cells, and cells co-expressing mCherry and BFP were sorted by FACSAria III cell sorter into 96-well plates to obtain single-cell-derived clones. To obtain 3xHA-TTP expressing cells, pLX303-SFFV-mCherry-P2A-3xHA-TTP was transduced into RAW264.7-Dox-Cas9 cells, and cells expressing mCherry were bulk sorted using a FACSAria III. Bone marrow-derived macrophages (BMDMs) were differentiated from bone marrow isolated from femurs and tibias of 8-to-12-week-old mice from Cas9 knock-in mice of both sexes (*Platt et al., 2014*). Femur and tibia marrow was centrifuged and cells were resuspended in DMEM. Cells were differentiated in DMEM (Sigma-Aldrich, D6429) containing recombinant M-CSF for 10 days. All cells were cultured at 37 °C and 5% $CO_2$ in a humidified incubator. All animals were maintained in the pathogen-free animal facility of the Research Institute of Molecular Pathology, and all procedures were carried out according to an ethical animal license that is approved and regularly controlled by the Austrian Veterinary Authorities (License Number: GZ: 516079/2017/14). All reagents used in this study are listed in the *Table 4*. All antibodies used in this study are listed in *Table 5*.

## Transfections

All transfections were perfomed by mixing DNA and polyethylenimine (PEI, Polysciences, 23966) in a 1:3 ratio (µg DNA/µg PEI) in DMEM (Sigma-Aldrich, D6429) without supplements. Transfection was performed using 500 ng of total DNA. The day before transfection, 2×10^5 HEK293T cells were seeded in six-well clusters in fully supplemented media. Cells were harvested 48 hr after transfection, washed with ice cold PBS and stored at −80 °C until further processing.

## Western blot

Cells were lysed in Frackelton lysis buffer (10 mM Tris-HCl pH 7.4, 50 mM NaCl, 30 mM $Na_4P_2O_7$, 50 mM NaF, 2 mM EDTA, 1% Triton X-100, 1 mM DTT, 0.1 mM PMSF, and 1 X protease inhibitor

**Table 1.** Vectors.

| Plasmid | Purpose | Reference or source |
| --- | --- | --- |
| pRRL-TRE3G-Cas9-P2A-GFP-PGK-IRES-rtTA3 | Dox inducible Cas9 | Johannes Zuber, IMP |
| pLX303-mCherry.TTP-P2A-BFP | TTP reporter | This study |
| pLX303-MYC-mCherry-P2A-3xHA.TTP | TTP reporter | This study |
| pLX303-MYC-mCherry-P2A-3xHA.TTP ΔC1 | TTP reporter | This study |
| pLX303-MYC-mCherry-P2A-3xHA.TTP ΔC2 | TTP reporter | This study |
| pLX303-MYC-mCherry-P2A-3xHA.TTP ΔC3 | TTP reporter | This study |
| pLX303-MYC-mCherry-P2A-3xHA.TTP ΔC4 | TTP reporter | This study |
| pLX303-MYC-mCherry-P2A-3xHA.TTP ΔC5 | TTP reporter | This study |
| pLX303-MYC-mCherry-P2A-3xHA.TTP ΔC6 | TTP reporter | This study |
| pLX303-MYC-mCherry-P2A-3xHA.TTP ΔN1 | TTP reporter | This study |
| pLX303-MYC-mCherry-P2A-3xHA.TTP ΔN2 | TTP reporter | This study |
| pLX303-MYC-mCherry-P2A-3xHA.TTP ΔN3 | TTP reporter | This study |
| pLX303-MYC-mCherry-P2A-3xHA.TTP ΔN4 | TTP reporter | This study |
| pLX303-MYC-mCherry-P2A-3xHA.TTP Δ259–278 | TTP reporter | This study |
| pLX303-MYC-mCherry-P2A-3xHA.TTP ΔC234-258 | TTP reporter | This study |
| pLX303-MYC-mCherry-P2A-3xHA.TTP Δ206–233 | TTP reporter | This study |
| pLX303-MYC-mCherry-P2A-3xHA.TTP KtoR | TTP reporter | This study |
| pLX303-MYC-mCherry-P2A-3xHA.TTP S52A S178A | TTP reporter | This study |
| pLX303-MYC-mCherry-P2A-3xHA.TTP ZNF C116R C139R | TTP reporter | This study |
| pLX303-MYC-mCherry-P2A-3xHA.TTP K97R | TTP reporter | This study |
| pLX303-MYC-mCherry-P2A-3xHA.TTP K115R | TTP reporter | This study |
| pLX303-MYC-mCherry-P2A-3xHA.TTP K133R | TTP reporter | This study |
| pLX303-MYC-mCherry-P2A-3xHA.TTP K135R | TTP reporter | This study |
| pLX303-MYC-mCherry-P2A-3xHA.TTP K141R | TTP reporter | This study |
| pLX303-MYC-mCherry-P2A-3xHA.TTP K97R/K115R | TTP reporter | This study |
| pLX303-MYC-mCherry-P2A-3xHA.TTP K97R/K115R/K133R | TTP reporter | This study |
| pLX303-MYC-mCherry-P2A-3xHA.TTP K97R/K115R/K133R/K135R | TTP reporter | This study |
| pLX303-MYC-mCherry-P2A-3xHA.TTP K135R/K141R | TTP reporter | This study |
| pLX303-MYC-mCherry-P2A-3xHA.TTP K133R/K135R/K141R | TTP reporter | This study |
| pLX303-MYC-mCherry-P2A-3xHA.TTP K115R/K133R/K135R/K141R | TTP reporter | This study |
| CMV-Flag-TTP | TTP reporter | Pavel Kovarik, Max Perutz Labs |
| DualCRISPR-hU6-sgRNA-mU6-sgRNA-EF1as-BFP | Dual sgRNA | de Almeida M, Hinterndorfer M et al, 2021 |
| pLentiv2-U6-PGK-iRFP670-P2A-Neo | Single sgRNA | de Almeida M, Hinterndorfer M et al, 2022 |
| pLentiv2-U6-PGK-BFP-P2A-Neo | Single sgRNA | de Almeida M, Hinterndorfer M et al, 2023 |
| PRRL-PBS-U6-sgRNA-EF1as-Thy1-P2A-NeoR (sgETN) | Library sgRNA | Johannes Zuber, IMP |

cocktail). Cells were incubated for 5 min on ice and then centrifuged at 18,500 x *g* for 10 min at 4 °C. The supernatant was transferred to a new tube and protein concentration was determined using Pierce BCA Protein Assay Kit (Thermo Fisher Scientific, 23225). Between 20 and 40 micrograms of protein were mixed with Laemmli sample buffer supplemented with β-mercaptoethanol and boiled

**Table 2.** sgRNA coding sequences.

| Gene | Species | Sequence (5' to 3') |
| --- | --- | --- |
| ROSA_1 (RAW264.7 & BMDMs) | mouse | AGATGGGCGGGAGTCTTC |
| ROSA_2 (RAW264.7 & BMDMs) | mouse | TTTAGATGGGCGGGAGTCTTCGTTTA |
| Huwe1_1 (RAW264.7 & BMDMs) | mouse | GATTTGCTGCAGTTCCAAG |
| Huwe1_2 (RAW264.7 & BMDMs) | mouse | ATAAAATTCAAAGTGTAGTG |
| Psmb7_1 (RAW264.7 & BMDMs) | mouse | GCTGTAACAACTCTCGGG |
| Psmb7_2 (RAW264.7 & BMDMs) | mouse | GAAAACTGGCACTACCATCG |
| Vcpip1 (RAW264.7 & BMDMs) | mouse | GACGTGCTCTGGTTCGATG |
| Ppil4 (RAW264.7 & BMDMs) | mouse | GTGTTTGGTGAAGTGACAGA |
| Tceb1 (RAW264.7 & BMDMs) | mouse | GCTGAGAATGAAACCAACG |
| Ube3c (RAW264.7 & BMDMs) | mouse | GAGAGTCAAAGTTCAAAA |
| Ddx23 (RAW264.7 & BMDMs) | mouse | GGATGGAGCGGGAGACCAA |
| Cnot10 (RAW264.7 & BMDMs) | mouse | GATTTCACAGGGTAGCGG |
| Ttp (RAW264.7 & BMDMs) | mouse | GAAGCGGGCGTTGTCGCTACG |
| AAVS1_1 (RKO & HEK-293T) | human | CTGTGCCCCGATGCACAC |
| AAVS1_2 (RKO & HEK-293T) | human | GCTGTGCCCCGATGCACAC |
| HUWE1_1 (RKO & HEK-293T) | human | GTGCGAGTTATATCACTGGG |
| HUWE1_2 (RKO & HEK-293T) | human | GTGCGAGTTATATCACTGGGTGG |
| AAVS1_3 (A375 & SW620) | human | GCTGTGCCCCGATGCACAC |
| AAVS1_4 (A375 & SW620) | human | GCTTGGCAAACTCACTCTT |
| HUWE1_3 (A375 & SW620) | human | GTGCGAGTTATATCACTGGG |
| HUWE1_4 (A375 & SW620) | human | GACAGTGGAGAATATGTCA |

for 10 min. Proteins were loaded on SDS polyacrylamide gels. The percentage of the gel was chosen based on the MW of the proteins of interest. Proteins were blotted on a PVDF or on a nitrocellulose membrane at 4 °C for 16 hr at 200 mA and then for 2 hr at 400 mA in Towbin buffer (25 mM Tris-HCl pH 8.3, 192 mM glycine, and 20% ethanol) or in carbonate transfer buffer (3 mM $Na_2CO_3$, 10 mM $NaHCO_3$, and 20% ethanol). The membrane was blocked in 5% BSA in PBS-T for 1 hr at room temperature and then incubated with the primary antibody overnight at 4 °C while shaking. The next day, the membrane was washed three times with PBS-T and incubated with HRP-coupled secondary antibody for 1 hr at room temperature and imaged with the ChemiDoc Imaging System from Bio-Rad. Relative protein levels were quantified using Image Lab (BioRad).

## Immunoprecipitation

Cells were lysed in 1 ml of RIPA lysis buffer (50 mM Tris-HCl pH 7.4, 150 mM NaCl, 1% SDS, 0.5% Sodium deoxycholate, 1% Triton X-100) supplemented with 40 mM NEM, 40 mM iodoacetamide, 25 U/ml Benzonase, 0.1 mM PMSF, and 1 X protease inhibitor. Cells were incubated on a rotating wheel at 4 °C for 30 min, and centrifuged at 20,000 x *g* at 4 °C for 30 min. The supernatant was transferred to a new tube and 50 µl (20% of the lysate used for the IP) were collected as input. A total of 500 µg of lysates were incubated overnight at 4 °C on a rotating wheel with an IgG Isotype control (Cell Signaling Technology, 1:300), anti-HA antibody (Cell Signaling Technology, 1:100), or anti-TTP antibody (Cell Signaling Technology, 1:100). The next day, magnetic beads (Pierce Protein A/G Magnetic Beads, Thermo Fisher Scientific, 88803) were blocked by rotation in 3% BSA in RIPA Buffer for 1 hr at 4 °C. Twenty-five µl of beads were added to 500 micrograms of lysates and rotated for 2 hr at 4 °C. Then, the beads were washed five times with 1 ml of RIPA buffer supplemented with

**Table 3.** Cells and culture conditions.

| Cell lines and primary cells | Type | Reference or source | Purpose | Media | Supplements |
|---|---|---|---|---|---|
| RAW264.7 | Murine macrophages | ATCC TIB-71 | parental cell line | Dulbecco's modified Eagle's medium (DMEM; Sigma-Aldrich, D6429) | 10% FBS (Sigma-Aldrich, F7524) and 1% penicillin/streptomycin (Sigma-Aldrich, P4333) |
| RAW264.7-Dox-Cas9 | Murine macrophages | This study | Dox-inducible Cas9 | Dulbecco's modified Eagle's medium (DMEM; Sigma-Aldrich, D6429) | 10% FBS (Sigma-Aldrich, F7524) and 1% penicillin/streptomycin (Sigma-Aldrich, P4333) |
| RAW264.7-Dox-Cas9 mCherry-TTP-P2A-BFP | Murine macrophages | This study | mCherry-TTP reporter | Dulbecco's modified Eagle's medium (DMEM; Sigma-Aldrich, D6429) | 10% FBS (Sigma-Aldrich, F7524) and 1% penicillin/streptomycin (Sigma-Aldrich, P4333) |
| RAW264.7-Dox-Cas9 mCherry-IkBα-P2A-BFP | Murine macrophages | This study | mCherry-IkBα reporter | Dulbecco's modified Eagle's medium (DMEM; Sigma-Aldrich, D6429) | 10% FBS (Sigma-Aldrich, F7524) and 1% penicillin/streptomycin (Sigma-Aldrich, P4333) |
| RAW264.7-Dox-Cas9 3xHA-TTP | Murine macrophages | This study | 3xHA-TTP reporter | Dulbecco's modified Eagle's medium (DMEM; Sigma-Aldrich, D6429) | 10% FBS (Sigma-Aldrich, F7524) and 1% penicillin/streptomycin (Sigma-Aldrich, P4333) |
| Bone Marrow Derived Macrophages, BMDMs | Murine macrophages | This study | constitutive Cas9 expression | Dulbecco's modified Eagle's medium (DMEM; Sigma-Aldrich, D6429) | 10% FBS (Sigma-Aldrich, F7524) and 1% penicillin/streptomycin (Sigma-Aldrich, P4333) |
| HEK293T | Human kidney neural tissue | CRL-3216 | 3xHA-TTP mutants | Dulbecco's modified Eagle's medium (DMEM; Sigma-Aldrich, D6429) | 10% FBS (Sigma-Aldrich, F7524) and 1% penicillin/streptomycin (Sigma-Aldrich, P4333) |
| Lenti-X 293T | Human kidney neural tissue | Takara, Cat# 632180 | VLP production | Dulbecco's modified Eagle's medium (DMEM; Sigma-Aldrich, D6429) | 10% FBS (Sigma-Aldrich, F7524) and 1% penicillin/streptomycin (Sigma-Aldrich, P4333) |
| RKO | human colon carcinoma | de Almeida M, Hinterndorfer M et al, 2021 | Dox-inducible Cas9 | RPMI 1640 (Thermo Fisher Scientific, 21875) | 10% FBS (Sigma-Aldrich, F7524), L-glutamine (4 mM, Gibco), sodium pyruvate (1 mM, Sigma-Aldrich), and 1% penicillin/streptomycin (Sigma-Aldrich, P4333) |

*Table 3 continued on next page*

*Table 3 continued*

| Cell lines and primary cells | Type | Reference or source | Purpose | Media | Supplements |
|---|---|---|---|---|---|
| A375 | human melanoma | This study | Dox-inducible Cas9 | Dulbecco's modified Eagle's medium (DMEM; Sigma-Aldrich, D6429) | 10% FBS (Sigma-Aldrich, F7524), L-glutamine (4 mM, Gibco) and 1% penicillin/streptomycin (Sigma-Aldrich, P4333) |
| SW620 | human colon carcinoma | This study | Dox-inducible Cas9 | Dulbecco's modified Eagle's medium (DMEM; Sigma-Aldrich, D6429) | 10% FBS (Sigma-Aldrich, F7524), L-glutamine (4 mM, Gibco) and 1% penicillin/streptomycin (Sigma-Aldrich, P4333) |

**Table 4.** reagents.

| Description | Abbreviation | Application | Dilution/concentration | Manufacturer | Catalogue number |
|---|---|---|---|---|---|
| Lipopolysaccharides from *Escherichia coli* O55:B5 | LPS | Cell culture | 10 ng/ml | Sigma-Aldrich | L2637 |
| Cycloheximide | CHX | Cell culture | 40 µg/m | Sigma-Aldrich | C1988 |
| MG132 | MG132 | Cell culture | 10 µM | Sigma-Aldrich | M7449 |
| Epoxomicin | EPX | Cell culture | 10 µM | Gentaur Molecular Products | 607-A2606 |
| TAK-243 | | Cell culture | 0.5 µM | ChemScence | CS-0019384 |
| Doxycycline hyclate | DOX | Cell culture | 500 ng/ml | Sigma-Aldrich | D9891 |
| G418 disulfate salt | G418 | Cell culture | 0.5–1 mg/ml | Sigma-Aldrich | A1720 |
| PH-797804, p38 inhibitor | p38i | Cell culture | 1 µM | Selleckchem | S2726 |
| JNK Inhibitor II, JNK inhibitor | JNKi | Cell culture | 20 µM | Sigma-Aldrich | 420119 |
| PF-3644022, MK2 inhibitor | MK2i | Cell culture | 10 µM | Sigma-Aldrich | PZ0188 |
| U0126, MEKi inhibitor | ERKi | Cell culture | 250 nM | Cell Signaling Technology | 9903 |
| Okadaic Acid | OA | Cell culture | 1 µM | Cell Signaling Technology | 5934 |
| Calyculin A | CalycA | Cell culture | 50 nM | Cell Signaling Technology | 9902 |
| Etoposide | | Cell culture | 5 µM | Sigma-Aldrich | E1383 |
| Brefeldin A | | Cell culture | 10 ug/ml | Sigma-Aldrich | B7651 |

**Table 5.** antibodies.

| Target | Application | Dilution | Conjugate | Manufacturer | Catalogue number | Name | Type |
|---|---|---|---|---|---|---|---|
| TTP | Western blot | 1:1000 | | Cell Signaling Technology | 71632 | D1I3T | Primary |
| Myc Tag | Western blot | 1:5000 | | Sigma-Aldrich | 05–724 | 4A6 | Primary |
| HA tag | Western blot | 1:1000 | | Cell Signaling Technology | 3724 | C29F4 | Primary |
| HECTH9 | Western blot | 1:1000 | | Cell Signaling Technology | 5695 | AX8D1 | Primary |
| Lasu1/Ureb1 | Western blot | 1:1000 | | Bethyl | A300-486A | | Primary |
| Vinculin | Western blot | 1:1000 | | Sigma-Aldrich | V9131 | V9131 | Primary |
| phospho-p38 MAPK, Thr180/Tyr182 | Western blot | 1:1000 | | Cell Signaling Technology | 9211 | | Primary |
| p38 MAPK | Western blot | 1:1000 | | Cell Signaling Technology | 9212 | | Primary |
| phospho-SAPK/JNK, Thr183/Tyr185 | Western blot | 1:1000 | | Cell Signaling Technology | 9251 | | Primary |
| SAPK/JNK | Western blot | 1:1000 | | Cell Signaling Technology | 9252 | | Primary |
| phospho-p44/42 MAPK (Erk1/2), Thr202/Tyr204 | Western blot | 1:1000 | | Cell Signaling Technology | 9101 | | Primary |
| p44/42 MAPK (Erk1/2) | Western blot | 1:1000 | | Cell Signaling Technology | 4695 | 137F5 | Primary |
| p-MK2 (Thr334) | Western blot | 1:1000 | | Cell Signaling Technology | 3007 | 27B7 | Primary |
| MK2 | Western blot | 1:1000 | | Cell Signaling Technology | 3042 | | Primary |
| p-p53, Ser15 | Western blot | 1:1000 | | Cell Signaling Technology | 9284 | | Primary |
| p-p53 | Western blot | 1:1000 | | Cell Signaling Technology | 2524 | 1C12 | Primary |
| ZFP36L1/2 | Western blot | 1:1000 | | Proteintech | 12306–1-AP | 12306–1-AP | Primary |
| Ubiquitin | Western blot | 1:1000 | | Santa Cruz Biotechnology | sc-8017 | P4D1 | Primary |
| HRP-β-actin | Western blot | 1:20000 | HRP | Abcam | ab49900 | AC-15 | Primary |
| HRP anti-rabbit IgG | Western blot | 1:3500 | HRP | Cell Signaling Technology | 7074 | | Secondary |
| HRP anti-mouse IgG | Western blot | 1:3500 | HRP | Cell Signaling Technology | 7076 | | Secondary |
| TTP | FACS | 1:100 | | Cell Signaling Technology | 71632 | D1I3T | Primary |
| HECTH9 | FACS | 1:100 | | Cell Signaling Technology | 5695 | AX8D1 | Primary |
| TNF alpha | FACS | 1:100 | APC | eBioscience | 17-7321-82 | MP6-XT22 | Primary |
| Rat IgG1 kappa Isotype Control | FACS | 1:500 | APC | eBioscience | 17-4301-82 | eBRG1 | Primary |
| APC anti-CD90.1/Thy1.1 | FACS | 1:500 | APC | BioLegend | 202526 | | Secondary |
| Alexa Fluor Plus 594 anti-Mouse IgG | FACS | 1:500 | Alexa Fluor 594 | Thermo Fisher Scientific | A-21201 | | Secondary |

*Table 5 continued on next page*

*Table 5 continued*

| Target | Application | Dilution | Conjugate | Manufacturer | Catalogue number | Name | Type |
|---|---|---|---|---|---|---|---|
| Alexa Fluor Plus 680 anti-Rabbit IgG | FACS | 1:500 | Alexa Fluor 680 | Thermo Fisher Scientific | A-21076 | | Secondary |
| APC anti-F4/80 | FACS | 1:100 | APC | Thermo Fisher Scientific | 17-4801-82 | BM8 | Secondary |
| TruStain FcX mouse Fc Receptor CD16/32 | FACS | 1:100 | | BioLegend | 101319 | | |
| IgG Isotype Control | IP | 1:300 | | Cell Signaling Technology | 2729 | | Primary |
| TTP | IP | 1:100 | | Cell Signaling Technology | 71632 | D1I3T | Primary |
| HA tag | IP | 1:100 | | Cell Signaling Technology | 3724 | C29F4 | Primary |

300 mM NaCl, and proteins were eluted in 2 X disruption buffer (2.1 M Urea, 667 mM β-mercaptoethanol and 1.4% SDS).

## Protein half-life determination

To estimate HA-TTP protein half-life, RAW264.7 Dox-Cas9 cells expressing sg*Huwe1* or sg*ROSA* were treated with Dox for 5 days before translational elongation was inhibited using 40 µg/ml of cycloheximide (CHX, Sigma-Aldrich, C1988). At indicated time points, whole cell lysates were prepared, analysed by western blot, quantified, and normalized to ACTIN levels and to time point 0 as indicated. Single exponential decay curves were determined using GraphPad Prism (v9), from which protein half-lives were calculated.

## RNA isolation, cDNA synthesis, and qPCR

Total RNA was extracted from mouse bone-marrow-derived macrophages and RAW264.7 Dox-Cas9 cells harboring non-targeting *ROSA* or *Huwe1*-targeting sgRNAs. $0.5 \times 10^6$ cells were lysed using Trizol reagent (Thermo-Fisher Scientific, 5596–018) and total RNA was isolated as recommended, and treated with 0.2 U/µl Turbo DNase (Thermo Fisher Scientific, AM2238). cDNA was prepared using Oligo (dT18) Primer (Thermo Fisher Scientific, S0132) or random hexamer primers (Thermo Fisher Scientific, S0142) and RevertAid Reverse Transcriptase (Thermo-Fisher Scientific, EP0441). Real-time PCR experiments were run on a Mastercycler (Biorad), using SYBR Green (Thermo-Fisher Scientific, S7567). Primers for qPCR are listed in *Table 6*.

**Table 6.** qPCR primers.

| Target | Primer | Sequence (5' to 3') |
|---|---|---|
| | FWD | CCAGAAACCGCTATGAAGTTCC |
| *Il6* | REV | TTGTCACCAGCATCAGTCCC |
| | FWD | CTCTGCCATCTACGAGAGCC |
| *Zfp36* | REV | GATGGAGTCCGAGTTTATGTTCC |
| | FWD | GATCGGTCCCCAAAGGGATG |
| *Tnf* | REV | CACTTGGTGGTTTGCTACGAC |
| | FWD | GGCAAAGAGGAACTGTAAG |
| *pre-Tnf* | REV | CCATAGAACTGATGAGAGG |
| | FWD | ATGGTGAAGGTCGGTGTGA |
| *Gapdh* | REV | TGAAGGGGTCGTTGATGG |

## Lentivirus production and transduction

Semiconfluent Lenti-X cells were transfected with mixes containing lentiviral transfer plasmids of interest, pCRV1-Gag-Pol (*Hatziioannou et al., 2004*) and pHCMV-VSV-G (*Yee et al., 1994*) using polyethylenimine (PEI, Polysciences, 23966) in a ratio of 1:3 (µg DNA/µg PEI) in DMEM without any supplements. Virus containing supernatant was clarified of cellular debris by filtration through a 0.45 µm filter. Virus-like particles were directly used after harvesting or kept at 4 °C for short-term storage. Target cells were infected in the presence of 6 µg/ml of polybrene (Sigma-Aldrich, TR1003G).

## Intracellular staining for flow cytometry

For staining of intracellular proteins, cells were collected and washed twice with PBS and subsequently fixed with 2% PFA for 15 min.at room temperature (RT). After PBS washes, cells were resuspended in ice-cold MeOH for permeabilization. At this point, fixed cells were stored in MeOH at −20 °C for a maximum of 2 days. On the day of the intracellular staining, cells were washed with PBS, and incubated for 10 min.at RT in TruStain FcX mouse Fc Receptor CD16/32 block to inhibit nonspecific antibody binding. Cells were then incubated with the primary antibody or left unstained for 1 hr at RT. Following three PBS washes, cells were incubated with the secondary antibody for 15 min at 4 °C. Cells were washed two times and resuspended in FACS buffer for flow cytometric analysis on an LSRFortessa (BD Biosciences) operated by BD FACSDiva software (v8.0). FACS data were analysed in FlowJo (v10.8). Median fluorescence intensities were normalized to the control.

## FACS-based CRISPR–Cas9 screens

The genome-wide Vienna sgRNA library was was lentivirally packaged in semiconfluent Lenti-X cells (Takara) via PEI transfection. Following double harvest, the collected supernatant was cleared of cellular debris by filtration through a 0.45 µm PES filter and stored a+4 °C. The obtained virus was used to transduce RAW264.7 Dox-Cas9 cells at a multiplicity of infection (MOI) of less than 0.2 TU/cell, and 600–1000-fold library representation. The percentage of library-positive cells was determined after 4 days of transduction by immunostaining of the Thy1.1 surface marker, and subsequent flow cytometric analysis. Library-positive cells were selected with G418 (1 mg/ml, Sigma-Aldrich, A1720) and expanded. Genome editing was induced with Dox (500 ng/ml, Sigma-Aldrich, D9891) and Cas9-GFP expression was monitored by FACS. Prior to Cas9 induction with Dox (Day 0), as well as before each FACS sort, an unsorted reference sample was collected. For this, a number of cells corresponding to at least 1000-fold library representation was collected and stored at −80 °C until further processing. After 3 days and 6 days of Cas9 induction, cells were sorted at FACS. Cells were harvested, washed with PBS and stained with Fixable Viability Dye eFluor (1:1,000, eBioscience, 65-0865-14) for 30 min. Subsequently, cells were washed three times with PBS, strained through a 35 µm nylon mesh and sorted in DMEM using the FACSAria II or FACSAria III cell sorters operated by BD FACSDiva software (v8.0). For the sort the following gating strategy was used: debris, doublets, dead (Viability Dye positive), Cas9-negative (GFP), mCherry- and BFP-negative cells were excluded. 5% of cells with the lowest and 1% of cells with the highest mCherry-TTP signal were sorted into PBS; same for the BFP internal control. At least $3×10^6$ (mCherry^low and BFP^low) and $5×10^5$ (mCherry^high and BFP^high) cells were collected for each time point. Sorted samples were re-analysed for purity, pelleted and stored at −80 °C until further processing. The gating strategy for flow cytometric cell sorting is shown in *Figure 2—source data 5*.

## Generation of next-generation sequencing libraries

Next-generation sequencing (NGS) libraries of sorted and unsorted control samples were processed as previously described (*de Almeida et al., 2021*). Isolated genomic DNA was subjected to two-step PCR. The first PCR allowed the amplification of the integrated sgRNA cassette, the second PCR introduced the Illumina adapters. Purified PCR products size distribution and concentration was measured using a fragment analyzer (Advanced Analytical Technologies). Equimolar ratios of the obtained libraries were pooled and sequenced on a HiSeq 2500 platform (Illumina). Primers used for library amplification are listed in *Table 7*. In primer sequences, NNNNNN denotes random nucleotides, XXXX denotes sample-specific barcodes.

## Analysis of pooled CRISPR screens

The analysis of the CRISPR–Cas9 screen was carried out as previously described (*de Almeida et al., 2021*). sgRNAs enriched in day 3 and day 6 sorted samples were calculated against the unsorted

**Table 7.** NGS library primers.

PCR 1

| Primer_name | Direction | Sequence | Comments |
|---|---|---|---|
| sgDeepSeq_rev_XXXX | Rv | CTCTTTCCCTACACGACGCTCTTCCGATCT NNNNNNCTCATTCCAGCATAGCTCTTAAAC | Library preparation 1st PCR |
| sgDeepSeq_rev_XXXX | Rv | CTCTTTCCCTACACGACGCTCTTCCGATCT NNNNNNTCGATTCCAGCATAGCTCTTAAAC | Library preparation 1st PCR |
| sgDeepSeq_rev_XXXX | Rv | CTCTTTCCCTACACGACGCTCTTCCGATCT NNNNNNCCTATTCCAGCATAGCTCTTAAAC | Library preparation 1st PCR |
| sgDeepSeq_rev_XXXX | Rv | CTCTTTCCCTACACGACGCTCTTCCGATCT NNNNNNGAACTTCCAGCATAGCTCTTAAAC | Library preparation 1st PCR |
| sgDeepSeq_rev_XXXX | Rv | CTCTTTCCCTACACGACGCTCTTCCGATCT NNNNNNATCCTTCCAGCATAGCTCTTAAAC | Library preparation 1st PCR |
| sgDeepSeq_rev_XXXX | Rv | CTCTTTCCCTACACGACGCTCTTCCGATCT NNNNNNACTCTTCCAGCATAGCTCTTAAAC | Library preparation 1st PCR |
| sgDeepSeq_rev_XXXX | Rv | CTCTTTCCCTACACGACGCTCTTCCGATCT NNNNNNCTTCTTCCAGCATAGCTCTTAAAC | Library preparation 1st PCR |
| sgDeepSeq_rev_XXXX | Rv | CTCTTTCCCTACACGACGCTCTTCCGATCT NNNNNNCAAGTTCCAGCATAGCTCTTAAAC | Library preparation 1st PCR |
| sgDeepSeq_rev_XXXX | Rv | CTCTTTCCCTACACGACGCTCTTCCGATCT NNNNNNTGAGTTCCAGCATAGCTCTTAAAC | Library preparation 1st PCR |
| sgDeepSeq_rev_XXXX | Rv | CTCTTTCCCTACACGACGCTCTTCCGATCT NNNNNNTTCGTTCCAGCATAGCTCTTAAAC | Library preparation 1st PCR |
| sgDeepSeq_rev_XXXX | Rv | CTCTTTCCCTACACGACGCTCTTCCGATCT NNNNNNTAGGTTCCAGCATAGCTCTTAAAC | Library preparation 1st PCR |
| sgDeepSeq_rev_XXXX | Rv | CTCTTTCCCTACACGACGCTCTTCCGATCT NNNNNNTCTGTTCCAGCATAGCTCTTAAAC | Library preparation 1st PCR |
| Fwd1_hybrid_P7_Nras | Fwd | GCATACGAGATAGCTAGCCACC | Library preparation 1st PCR |

PCR 2

| Primer_name | Direction | Sequence | Comments |
|---|---|---|---|
| Rev2_p5_sgDeepSeq | Rv | AATGATACGGCGACCACCGAGATCTACACT CTTTCCCTACACGACGCT | Library preparation 2nd PCR |
| Fwd2_p7_sgDeepSeq | Fwd | CAAGCAGAAGACGGCATACGAGATAGCTAGCCACC | Library preparation 2nd PCR |

population control harvested on the respective day of sorting. In addition, the dropout over time of sgRNAs was calculated by comparing the unsorted populations to the initial day 0 population.

## Quantitative proteomics

To systematically assess protein changes after *Huwe1* or *Psmb7* knockout, in RAW264.7-Dox-Cas9 cells *ROSA*-, *Huwe1*- and *Psmb7*-KO was induced for 2 days. Cells where incubated with LPS for the indicated times, after which $5 \times 10^5$ cells were washed with PBS, pelleted and snap-frozen and stored at −80 °C. Sample protein concentration was measured using standard Pierce Protein Assay Kit (Thermo Fisher Scientific, 23225), after which 40 µg of protein were processed using the iST PreOmics Sample Preparation kit 8 x (P.O. 00001) according to the manufacturer's instructions.

Peptides were separated on an Ultimate 3000 RSLC nano-flow chromatography system (Thermo Fisher Scientific), using a pre-column for sample loading (Acclaim PepMap C18, 2 cm ×0.1 mm, 5 µm, Thermo Fisher Scientific), and a C18 analytical column (Acclaim PepMap C18, 50 cm ×0.75 mm, 2 µm, Thermo Fisher Scientific), applying a segmented linear gradient from 2% to 35% and finally 80% solvent B (80% acetonitrile, 0.1% formic acid; solvent A 0.1% formic acid) at a flow rate of 230 nL/min over 120 min. Eluting peptides were analyzed on an Exploris 480 Orbitrap mass spectrometer (Thermo

Fisher Scientific), which was coupled to the column with a FAIMS pro ion-source (Thermo Fisher Scientific) using coated emitter tips (PepSep, MSWil). The mass spectrometer was operated in DIA mode with the FAIMS CV set to –45, the survey scans were obtained in a mass range of 400–900 m/z, at a resolution of 120 k at 200 m/z and a normalized AGC target at 300%. 31 MS/MS spectra with variable isolation width between 13 and 24 m/z covering 399.5–899.5 m/z range including 1 m/z windows overlap, were acquired in the HCD cell at 30% collision energy at a normalized AGC target of 1000% and a resolution of 30 k. The max. injection time was set to auto.

Raw data were processed using Spectronaut software (version 15.4.210913.50606, https://biognosys.com/software/spectronaut/) with the DirectDIA workflow. The Uniprot mouse reference proteome (version 2021.03, https://www.uniprot.org), as well as a database of most common contaminants were used. The searches were performed with full trypsin specificity and a maximum of 2 missed cleavages at a protein and peptide spectrum match false discovery rate of 1%. Carbamidomethylation of cysteine residues were set as fixed, oxidation of methionine and N-terminal acetylation as variable modifications. The global normalization and imputation were done in Spectronaut - all other parameters were left at default. Spectronaut output tables were further processed using Cassiopeia_LFQ in R (https://github.com/moritzmadern/Cassiopeia_LFQ; *Madern, 2021*). Contaminant proteins, protein groups identified only by one peptide and protein groups with less than two quantitative values in one experimental group, were removed for further analysis. Differences between groups were statistically evaluated using the LIMMA package (*Ritchie et al., 2015*) in Cassiopeia_LFQ at 5% FDR (Benjamini-Hochberg).

To generate A375 AAVS1- and HUWE1-KO cells, inducible Cas9 clones of both cell lines were transduced with an sgRNA construct targeting the respective gene by lentiviral delivery. Cells were antibiotic selected for genomic integration and expanded. A375 samples were further FACS sorted for sgRNA + cells to obtain purity of >99%. Cells were harvested 72 hr after Cas9 induction with Dox at final concentration of 0.2 µg/µl. A total number of 3×10$^6$ SW620 or 2.5×10$^6$ A375 cells were harvested, washed with PBS, pelleted, snap-frozen, and stored at –70 °C. Protein concentrations were measured using standard BCA assay and normalized to 50 micrograms. The protein samples were prepared with iST preOmics Sample Preparation kit 96 x (P.O. 00027) according to the manufacturer's protocol.

The nano HPLC system (UltiMate 3000 RSLC nano system, Thermo Fisher Scientific) was coupled to an Orbitrap Eclipse Tribrid mass spectrometer equipped with a FAIMS pro interfaces and a Nanospray Flex ion source (all parts Thermo Fisher Scientific). Peptides were loaded onto a trap column (PepMap Acclaim C18, 5 mm ×300 µm ID, 5 µm particles, 100 Å pore size, Thermo Fisher Scientific) at a flow rate of 25 µl/min. using 0.1% TFA as mobile phase. After 10 minutes, the trap column was switched in line with the analytical column (PepMap Acclaim C18, 500 mm ×75 µm ID, 2 µm, 100 Å, Thermo Fisher Scientific) operated at 30 °C. Peptides were eluted using a flow rate of 230 nl/min, starting with the mobile phases 98% A (0.1% formic acid in water) and 2% B (80% acetonitrile, 0.1% formic acid) and linearly increasing to 35% B over the next 180 minutes. The Eclipse was operated in data-dependent mode, performing a full scan (m/z range 350–1500, resolution 120,000, target value 1E6) at 4 different compensation voltages (CV-45,–55, –65,–75), followed each by MS/MS scans of the most abundant ions for a cycle time of 0.75 sec per CV. MS/MS spectra were acquired using an isolation width of 1.2 m/z, target value of 3E4 and intensity threshold of 5E4, maximum injection time 20ms, HCD with a collision energy of 30, using the Iontrap for detection in the rapid scan mode. Precursor ions selected for fragmentation (include charge state 2–6) were excluded for 60 s. The monoisotopic precursor selection filter and exclude isotopes feature were enabled.

For peptide identification, the RAW-files were loaded into Proteome Discoverer (version 2.5.0.400, Thermo Fisher Scientific). All MS/MS spectra were searched using MSAmanda v2.0.0.16129 (*Dorfer et al., 2014*). The peptide mass tolerance was set to ±10 ppm, the fragment mass tolerance to ±400 mmu, the maximal number of missed cleavages was set to 2, using tryptic enzymatic specificity without proline restriction. Peptide and protein identification were performed in two steps. For an initial search, the RAW-files were searched against the database human_uniprot_reference_2021-06-30.fasta (20,531 sequences; 11,395,157 residues), supplemented with common contaminants, using the following search parameters: alkylation of cysteine by C6H11NO was set as a fixed modification, oxidation of methionine as variable modification. The result was filtered to 1% FDR on protein level using the Percolator algorithm (*Käll et al., 2007*) integrated in Proteome Discoverer. A

sub-database of proteins identified in this search was generated for further processing. For the second search, the RAW-files were searched against the created sub-database using the same settings as above plus considering additional variable modifications: phosphorylation on serine, threonine and tyrosine, deamidation on asparagine and glutamine, and glutamine to pyro-glutamate conversion at peptide N-terminal glutamine, acetylation on protein N-terminus were set as variable modifications. The localization of the post-translational modification sites within the peptides was performed with the tool ptmRS, based on the tool phosphoRS (*Taus et al., 2011*). Identifications were filtered again to 1% FDR on protein and PSM level, additionally an Amanda score cut-off of at least 70 was applied. Peptides were subjected to label-free quantification using IMP-apQuant (*Doblmann et al., 2019*). Proteins were quantified by summing unique and razor peptides and applying intensity-based absolute quantification (iBAQ; *Schwanhäusser et al., 2011*) with subsequent normalisation based on the MaxLFQ algorithm (*Cox et al., 2014*). Proteins were filtered to be identified by a minimum of 3 quantified in at least 1 sample. Protein-abundance-normalization was done using sum normalization. Statistical significance of differentially expressed proteins was determined using limma (*Smyth, 2004*).

## Acknowledgements

We are grateful to Johannes Bock for establishing reagents and methodology that enabled this work. We thank Kitti Csalyi, Johanna Stranner, and Thomas Sauer at the Max Perutz Labs BioOptics FACS Facility for expert support, Markus Hartl and the Max Perutz Labs Mass Spectrometry Facility for mass spectrometry analysis, the IMP/IMBA Protein Biochemistry Core Facility for performing quantitative proteomics, and the Vienna Biocenter Core Facilities (VBCF) for Next Generation Sequencing analysis. We thank Laura Boccuni for expert advice, and the 'Signaling Mechanisms in Cellular Homeostasis' doctoral program community, Manuela Baccarini, Thomas Decker, Pavel Kovarik and their labs for their technical expertise and help. *Funding sources:* This work was funded by Stand-Alone grants (P30231-B, P30415-B), Special Research Grant (SFB grant F79), and Doctoral School grant (DK grant W1261) from the Austrian Science Fund (FWF) to GAV, FWF grants P33000-B, P31848-B, and W1261 to PK, a Starting Grant from the European Research Council (ERC-StG-336860) to JZ, the Austrian Science Fund (SFB grant F4710) to JZ, Austrian Science Fund Special Research Grant (FWF, SFB F 79) to TC, and an ERC European Union's Horizon 2020 research and innovation programme grant (AdG 694978) to TC. SS is the recipient of a DOC fellowship of the Austrian Academy of Sciences. MS is a member of the Boehringer Ingelheim Discovery Research global post-doc program. Research at the IMP is supported by Boehringer Ingelheim and the Austrian Research Promotion Agency (Headquarter grant FFG-852936).

## Additional information

### Funding

| Funder | Grant reference number | Author |
|---|---|---|
| Austrian Science Fund | P30231-B | Gijs A Versteeg |
| Austrian Science Fund | P30415-B | Gijs A Versteeg |
| Austrian Science Fund | W1261 | Pavel Kovarik |
| Austrian Science Fund | P33000-B | Pavel Kovarik |
| Austrian Science Fund | P31848-B | Pavel Kovarik |
| European Research Council | ERC-StG-336860 | Johannes Zuber |
| Austrian Science Fund | F4710 | Tim Clausen |
| Austrian Science Fund | F79 | Tim Clausen |
| European Research Council | AdG 694978 | Tim Clausen |

| Funder | Grant reference number | Author |
|---|---|---|

The funders had no role in study design, data collection and interpretation, or the decision to submit the work for publication.

## Author contributions

Sara Scinicariello, Conceptualization, Formal analysis, Validation, Investigation, Visualization, Methodology, Writing – original draft, Project administration, Writing – review and editing; Adrian Soderholm, Formal analysis, Investigation, Methodology, Writing – review and editing; Markus Schäfer, Conceptualization, Methodology, Writing – review and editing; Alexandra Shulkina, Formal analysis, Investigation, Methodology; Irene Schwartz, Robert Kalis, Investigation, Methodology, Writing – review and editing; Kathrin Hacker, Valentina Budroni, Investigation, Methodology; Rebeca Gogova, Investigation, Writing – review and editing; Kimon Froussios, Formal analysis; Annika Bestehorn, Tim Clausen, Pavel Kovarik, Johannes Zuber, Conceptualization, Writing – review and editing; Gijs A Versteeg, Conceptualization, Supervision, Funding acquisition, Writing – original draft, Writing – review and editing

## Author ORCIDs

Robert Kalis ⬤ http://orcid.org/0000-0001-7553-4806
Kimon Froussios ⬤ http://orcid.org/0000-0003-2812-0525
Tim Clausen ⬤ http://orcid.org/0000-0003-1582-6924
Johannes Zuber ⬤ http://orcid.org/0000-0001-8810-6835
Gijs A Versteeg ⬤ http://orcid.org/0000-0002-6150-2165

## Ethics

All animals were maintained in the pathogen-free animal facility of the Research Institute of Molecular Pathology, and all procedures were carried out according to an ethical animal license that is approved and regularly controlled by the Austrian Veterinary Authorities (License Number: GZ: 516079/2017/14).

## Decision letter and Author response

Decision letter https://doi.org/10.7554/eLife.83159.sa1
Author response https://doi.org/10.7554/eLife.83159.sa2

---

# Additional files

## Supplementary files

• MDAR checklist

## Data availability

The mass spectrometry proteomics data have been deposited to the ProteomeXchange Consortium via the PRIDE partner repository with the dataset identifiers PXD036714, PXD036715, and PXD038892. All other data generated or analyzed during this study are included in the manuscript and supporting files.

The following datasets were generated:

| Author(s) | Year | Dataset title | Dataset URL | Database and Identifier |
|---|---|---|---|---|
| Versteeg GA | 2023 | HUWE1 controls tristetraprolin proteasomal degradation by regulating its phosphorylation | https://www.ebi.ac.uk/pride/archive/projects/PXD036714 | PRIDE, PXD036714 |
| Versteeg GA | 2023 | HUWE1 controls tristetraprolin proteasomal degradation by regulating its phosphorylation | https://www.ebi.ac.uk/pride/archive/projects/PXD036715 | PRIDE, PXD036715 |
| Versteeg GA | 2023 | HUWE1 controls tristetraprolin proteasomal degradation by regulating its phosphorylation | https://www.ebi.ac.uk/pride/archive/projects/PXD038892 | PRIDE, PXD038892 |

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
