## [Editor Report]

The study by Scinicariello et al. set out to identify novel factors that controlled TTP stability and identified HUWE1 by CRISPR screening in macrophages. HUWE1 limited TTP phosphorylation at later times post inflammatory stimulation on residues distinct from those functionally modified by MAPKs in early stages. Overall, the biochemical and cellular signaling experiments were thoughtfully designed and well executed, leading to the discovery of HUWE1 as a TTP regulator.

---

## [Decision Letter]

**Decision letter after peer review:**

Thank you for submitting your article "HUWE1 controls tristetraprolin proteasomal degradation by regulating its phosphorylation" for consideration by *eLife*. Your article has been reviewed by 3 peer reviewers, one of whom is a member of our Board of Reviewing Editors, and the evaluation has been overseen by David Ron as the Senior Editor. The reviewers have opted to remain anonymous.

Essential revisions:

Please experimentally address the questions raised by the reviewers.

*Reviewer #1 (Recommendations for the authors):*

The study by Scinicariello et al. set out to identify novel factors that controlled TTP stability and identified HUWE1 by CRISPR screening in macrophages. HUWE1 phosphorylated TTP on residues distinct from those phosphorylated by MAPKs and regulated TTP protein stability. Overall, the biochemical and cellular signaling experiments were thoughtfully designed and well executed, leading to the discovery of HUWE1 as a TTP regulator. Addressing the following questions may help strengthen the conclusion that HUWE1 regulates macrophage responses at the post-transcriptional steps.

1. In Figure 3B, the authors showed that HUWE1-deficient BMDMs expressed reduced levels of Il6 mRNA upon LPS stimulation. It would be desirable for the authors to assess the levels of Il6 primary transcripts, a proxy of Il6 gene transcription, to exclude the possibility that HUWE1 affected mRNA synthesis instead of TTP-mediated degradation.

2. It would also be desirable to measure protein levels of prototypical inflammatory cytokines in HUWE1-deficient BMDMs to ascertain the effects of HUWE1 on inflammatory outcomes.

3. For some reason, the resolution of Figure 7 is low such that the model could not be properly visualized and interpreted.

*Reviewer #2 (Recommendations for the authors):*

Sara Scinicariello et al., found that ubiquitin E3 ligase HUWE1 regulated proteasomal degradation of TTP by regulating its phosphorylation. Their conclusions are supported by biochemistry experiments. However, the physiological importance of the process was not investigated here. Moreover, the accurate mechanism by which HUWE1 regulated the phosphorylation of TTP was not clarified.

Specific points:

1. Authors indicated that LPS-induced TPP hyperphosphorylation in Figure 1B and 1G, but the result in Figure 1F was not as significant as in Figure 1B and 1G.

2. In Figure 2C, the expression of Endogenic TPP was up-regulated after LPS stimulation, but the expression of mCherry TTP was down-regulated. Contradiction with the trend mentioned by the author in Figure 1E.

3. The image of all figures' quality and resolution is very low, and it is difficult to read and understand.

4. Line 209: "either result in increased intracellular TTP protein concentrations ". This sentence looks redundant with "increased TTP protein levels upon Huwe1 ablation"

5. In figure 3, the author examined the induction of IL6 and TNF. As the experiments were performed in WT and HUWE1 KO (or KD) cell lines. It is possible that HUWE1 might regulate the induction of these cytokines. Control should be added before drawing a conclusion. For example, reconstitute TTP into HUWE1 KO (or KD) cell line.

6. Line 237~238: It is weird that the association between TTP and HUWE1 was not detected with co-IP or turbo-ID assay. As turbo-ID was able to detect weak and transient interaction.

7. Line 316: In figure 4E, Since both sets have no significant difference between control and Huwe1 KO, it is not proper to conclude a "rescue", especially with nearly the same error bar.

8. The authors should identify the exact AA sites where HUWE1 affects TTP ubiquitination and phosphorylation and performed mutation validation.

9. AA 234-258 are almost all phosphorylated regions of TTP (Figure 6C), deletion of the 234-258 region of TTP fails to hyperphosphorylate in HUWE1 knockout cells and cannot be proved to be the effect of HEWE1.

10. HUWE1 acts as an E3 ubiquitin ligase and has no interaction with TTP. How HUWE1 affects the ubiquitination and phosphorylation of TTP. The author should explain this.

11. The authors did not provide enough evidence to support that HUWE1 phosphorylates TTP via PP1/2.

*Reviewer #3 (Recommendations for the authors):*

Figure 1A: Mg132 treatment reveals proteasome-dependent degradation of endogenous TTP. Sensitivity to TAK243 should also be assessed to determine whether, as observed with the overexpressed TTP-HA protein, endogenous TTP degradation is sensitive to ubiquitination inhibitors.

Similarly, the claim that TTP is degraded in a ubiquitin-dependent manner (see the title of Figure 1) should be supported by data demonstrating that endogenous TTP protein is modified by ubiquitination. This could be achieved by an experiment similar to that shown in 1C performed by immunoprecipitation of endogenous TTP protein. Alternatively, a new approach to identify endogenous ubiquitination could be tested (see for example Zhang et al. PLOS Biology 2022 20:e3001501).

Sup. Figure 1B: Half-lifes of TTP wt and TTP K→R mutant should be compared by measuring the decrease in protein level after translational blockade (as performed in Figure 1A and 2G) to convincingly demonstrate that the absence of ubiquitination is correlated with the stabilization of TTP.

Figure 1E: In order to demonstrate a specific variation of mCherry-TTP level in response to LPS (similar to what is observed for the endogenous TTP protein), the level of mCherry-TTP measured in experiment 1E should be expressed relative to the level of eBFP. This would normalize potential variations in lentiviral promoter activity under the effect of LPS treatment.

---

## [Author Response]

Reviewer #1 (Recommendations for the authors):1. In Figure 3B, the authors showed that HUWE1-deficient BMDMs expressed reduced levels of Il6 mRNA upon LPS stimulation. It would be desirable for the authors to assess the levels of Il6 primary transcripts, a proxy of Il6 gene transcription, to exclude the possibility that HUWE1 affected mRNA synthesis instead of TTP-mediated degradation.

To determine whether *Huwe1* knock-out influences TTP-target mRNAs at a posttranscriptional level, we analyzed the mature and pre-mRNA levels of one of TTP’s bestcharacterized targets -*Tnf-* by RT-qPCR using exon-exon, and intron-exon primer pairs, respectively. These new data have been added as Figure 3A and 3B in the revised manuscript.

Consistent with the effects of *Huwe1*-loss on TTP-regulated mRNAs stemming from post-transcriptional effects, *Tnf* pre-mRNA levels were unaffected in *Huwe1* KO cells, whereas its mature mRNA was decreased (consistent with increased intra-cellular levels of biologically active TTP protein).

2. It would also be desirable to measure protein levels of prototypical inflammatory cytokines in HUWE1-deficient BMDMs to ascertain the effects of HUWE1 on inflammatory outcomes.

To determine whether differences in mRNA levels in *Huwe1* KO cells are reflected at the protein level, intra-cellular TNF protein abundance was determined by flow-cytometry. New data have been added as Figure 3 —figure supplement 1F in the revised manuscript.

Consistent with our *Tnf* mRNA data, there were no significant changes in intra-cellular TNF protein *HUWE1* KO cells at 3 h post LPS stimulation. However, as is the case for its mRNA, at later time points post-stimulation (6, and 9 h), TNF protein was significantly reduced in the absence of *Huwe1*.

3. For some reason, the resolution of Figure 7 is low such that the model could not be properly visualized and interpreted.

This likely happened during PDF conversion of the original manuscript. The revised PDF manuscript figures, and uploaded individual figures should be high resolution now, and allow for proper evaluation.

Reviewer #2 (Recommendations for the authors):Sara Scinicariello et al., found that ubiquitin E3 ligase HUWE1 regulated proteasomal degradation of TTP by regulating its phosphorylation. Their conclusions are supported by biochemistry experiments. However, the physiological importance of the process was not investigated here. Moreover, the accurate mechanism by which HUWE1 regulated the phosphorylation of TTP was not clarified.Specific points:1. Authors indicated that LPS-induced TPP hyperphosphorylation in Figure 1B and 1G, but the result in Figure 1F was not as significant as in Figure 1B and 1G.

We acknowledge that there are some minor differences in the magnitude of phosphorylation, and phosphorylation timing/kinetics between mCherry-TTP, HA-TTP, and endogenous TTP. Nevertheless, in all cases, all constructs in all tested cell types behaved conceptually in a similar manner. Specifically, we interpreted endogenous TTP to be substantially phosphorylated upon LPS treatment in Figure 1F, as phosphatase treatment collapsed the higher MW TTP species. The same was the case for mCherry-TTP upon phosphatase treatment. The increase in mCherry-TTP MW through LPS treatment is less pronounced on the blot, which we interpreted to stem from the more limited size-resolution higher up in the SDS-PAGE gel.

2. In Figure 2C, the expression of Endogenic TPP was up-regulated after LPS stimulation, but the expression of mCherry TTP was down-regulated. Contradiction with the trend mentioned by the author in Figure 1E.

We do not think that the data from these two panels in contradictory. The data presented in Figure 1E show that mCherry-TTP is stabilized at earlier times post-LPS stimulation (peak at 3 h), yet decreased below the initial steady-state levels by 6 h post-LPS stimulation. Collectively, our data suggest that this behavior stems from initial LPS-induced TTP stabilization (at 3 h), followed by LPS-dependent TTP degradation (≥6 h).

In order to measure the effect of *Huwe1*-loss on TTP degradation, the experiment in Figure 2C was performed at 6 h post-LPS stimulation; i.e. at a time during which mCherry-TTP is already lower than its initial steady-state levels in the sg*ROSA*+LPS control samples. Therefore, our conclusion is that the results in Figure 1E and 2C are in fact consistent.

3. The image of all figures' quality and resolution is very low, and it is difficult to read and understand.

This likely happened during PDF conversion of the original manuscript. The revised PDF manuscript figures, and uploaded individual figures should be high resolution now, and allow for proper evaluation.

4. Line 209: "either result in increased intracellular TTP protein concentrations ". This sentence looks redundant with "increased TTP protein levels upon Huwe1 ablation"

We agree, and have adapted the revised manuscript text by removing “result in increased intracellular TTP protein concentrations, and”.

5. In figure 3, the author examined the induction of IL6 and TNF. As the experiments were performed in WT and HUWE1 KO (or KD) cell lines. It is possible that HUWE1 might regulate the induction of these cytokines. Control should be added before drawing a conclusion. For example, reconstitute TTP into HUWE1 KO (or KD) cell line.

To determine whether *Huwe1* knock-out influences TTP-target mRNAs at a posttranscriptional level, we analyzed the mature and pre-mRNA levels of one of TTP’s bestcharacterized targets -*Tnf-* by RT-qPCR using exon-exon, and intron-exon primer pairs, respectively. These new data have been added as Figure 3A and 3B in the revised manuscript.

Consistent with the effects of *Huwe1*-loss on TTP-regulated mRNAs stemming from post-transcriptional effects, *Tnf* pre-mRNA levels were unaffected in *Huwe1* KO cells, whereas its mature mRNA was decreased (consistent with increased intra-cellular levels of biologically active TTP protein).

6. Line 237~238: It is weird that the association between TTP and HUWE1 was not detected with co-IP or turbo-ID assay. As turbo-ID was able to detect weak and transient interaction.

We concluded from these results that there are two possible explanations: (1) TTP is a direct substrate for HUWE1, resulting in its poly-ubiquitination and degradation, but the HUWE1-TTP interaction in cells is too weak to measure even by TurboID, or (2) HUWE1 affects TTP stability indirectly, e.g. by altering its phosphorylation.

Our data cannot exclude option 1 to be true (i.e. that HUWE1 ubiquitinates TTP). However, irrespective of whether option 1 is true, data presented in Figure 4 indicate that HUWE1 (also) affects TTP stability indirectly, by regulating its phosphorylation. Text in lines 249-259 of the revised manuscript addresses these points.

7. Line 316: In figure 4E, Since both sets have no significant difference between control and Huwe1 KO, it is not proper to conclude a "rescue", especially with nearly the same error bar.

We agree, and updated the text in the revised manuscript (lines 329-336).

8. The authors should identify the exact AA sites where HUWE1 affects TTP ubiquitination and phosphorylation and performed mutation validation.

The strong hyper-phosphorylation of TTP in the absence of HUWE1 indicates that many sites on TTP are (de)phosphorylated in a HUWE1-dependent manner, and thus likely contribute to regulation of TTP turnover. We aim to systematically test the contribution of each phospho-site to TTP stability regulation in the future, yet are of the opinion that this is not essential to the claims made in the current manuscript.

Moreover, we performed additional single and combinatorial KtoR mutagenesis. New data from the single mutants was added as Figure 1 —figure supplement 1F. TTP became fully stable only when all five lysine residues were muted (as shown in the original manuscript). This indicates that multiple lysine residues are ubiquitinated, and that single mutants can be compensated for by the other residues. This is not uncommon for many E3s, which have the structural flexibility to ubiquitinate any residues in particular substrate surfaces.

Our data indicate that HUWE1-related phosphorylation and ubiquitination events occur on multiple sites in TTP, and are functionally redundant/complementary.

9. AA 234-258 are almost all phosphorylated regions of TTP (Figure 6C), deletion of the 234-258 region of TTP fails to hyperphosphorylate in HUWE1 knockout cells and cannot be proved to be the effect of HEWE1.

Since the 234-258 region is not presented in Figure 6C, we assumed that Figure 6E was meant instead. Moreover, it is not clear to us exactly what the reviewer meant here, as deletion of the 234-258 region resulted in more TTP phosphorylation in sg*AAVS1* control cells (Figure 6E, and Figure 6 —figure supplement 1G), and this was not affected by *Huwe1* KO.

Nevertheless, from Figure 6E we concluded that the 234-258 is a likely binding site for the TTP phosphatase. We interpreted these data as follows: (1) in the absence of phosphatase binding (i.e. 234-258 deletion), TTP is hyper-phosphorylated, and (2) since *Huwe1* KO no longer increased TTP abundance of this mutant, the effect of HUWE1 on wild-type TTP occurs -at least in part- through differential TTP phosphorylation.

10. HUWE1 acts as an E3 ubiquitin ligase and has no interaction with TTP. How HUWE1 affects the ubiquitination and phosphorylation of TTP. The author should explain this.

This is discussed in detail in revised manuscript lines 468-479 (Results section) and 569-581 (Discussion section), and illustrated in Figure 7.

11. The authors did not provide enough evidence to support that HUWE1 phosphorylates TTP via PP1/2.

In our opinion, we do not make any strong claims in the manuscript that HUWE1 regulates PP1/2, rather indicate that aberrant PP1/2 activity would offer a reasonable explanation for the *Huwe1* KO phenotype.

In our opinion, the data presented in Figures 4E-F demonstrate with two independent sets of inhibitors, that prevent either phosphorylation, or dephosphorylation have the same effect: it negates the increase in TTP abundance caused by loss of *Huwe1*. Our main conclusion from this is, that the *Huwe1* KO phenotype is dependent on TTP phosphorylation, and that either preventing TTP phosphorylation (4i), or saturating it (CalycA), renders TTP insensitive to *Huwe1* loss.

Based on (1) aberrant activation of multiple stress kinases (Figure 4A-D), (2) these inhibitor experiments in Figure 4E-F, and (3) independent mutagenesis experiments in Figure 6, we *suggest* that the HUWE1-dependent effects on TTP *may* be through PP1/2. We are aiming to address this in more detail in future experiments, outside of the scope of this manuscript.

Reviewer #3 (Recommendations for the authors):Figure 1A: Mg132 treatment reveals proteasome-dependent degradation of endogenous TTP. Sensitivity to TAK243 should also be assessed to determine whether, as observed with the overexpressed TTP-HA protein, endogenous TTP degradation is sensitive to ubiquitination inhibitors.

New data assessing the effect of TAK-243 on endogenous TTP was added as Figure 1 —figure supplement 1B. Consistent with our data with exogenously expressed TTP, treatment with the inhibitor increased the abundance of endogenous TTP.

Similarly, the claim that TTP is degraded in a ubiquitin-dependent manner (see the title of Figure 1) should be supported by data demonstrating that endogenous TTP protein is modified by ubiquitination. This could be achieved by an experiment similar to that shown in 1C performed by immunoprecipitation of endogenous TTP protein. Alternatively, a new approach to identify endogenous ubiquitination could be tested (see for example Zhang et al. PLOS Biology 2022 20:e3001501).

New data assessing the ubiquitination of endogenous TTP was added as Figure 1 —figure supplement 1D.

Sup. Figure 1B: Half-lifes of TTP wt and TTP K→R mutant should be compared by measuring the decrease in protein level after translational blockade (as performed in Figure 1A and 2G) to convincingly demonstrate that the absence of ubiquitination is correlated with the stabilization of TTP.

New data from TTP-KtoR chase experiments was added as Figure 1 —figure supplement 1E. The half-life was increased substantially from 1.4 h for wtTTP to 5.7 h for the mutant.

Figure 1E: In order to demonstrate a specific variation of mCherry-TTP level in response to LPS (similar to what is observed for the endogenous TTP protein), the level of mCherry-TTP measured in experiment 1E should be expressed relative to the level of eBFP. This would normalize potential variations in lentiviral promoter activity under the effect of LPS treatment.

All of our FACS-based experiments with the mCherry-TTP-P2A-eBFP2 cells (e.g. Figure 2 —figure supplement 2E) have consistently demonstrated that BFP levels are not affected by *Huwe1* KO. Since the data from Figure 1E come from Western blot analyses, BFP values were not measured, and could not be used for normalization.